## RESEARCH ARTICLE

# Differentially expressed fusogens specify myocyte states to drive myogenesis

Sarah Nahlé[1,2], Awais Javed[2,3], Loïck Joumier[2,4], Yacine Kherdjemil[2], Julie Sitolle[5], Konstantin Khetchoumian[2], Yash Parekh[6], Wojciech Krezel[6], Mohan Malleshaiah[1,2,4], Fabien Le Grand[5], Michel Cayouette[1,2,7,8] and Jean-François Côté[1,2,4,7,8,*]

## ABSTRACT

During myogenesis, myocyte fusion leads to the formation of multinucleated muscle fibers, but how exactly this process is initiated remains poorly understood. Here, we performed single-cell RNA-sequencing on mouse somites from E9.5-E11.5 embryos, revealing multiple differentiation states during primary myogenesis. Among these, we identified two unexpected myocyte populations: one expressing both myomaker (*Mymk*) and myomixer (*Mymx*) (termed Mc1) and another expressing only *Mymk* (termed Mc2). Fluorescence *in situ* hybridization demonstrated that both populations are mononucleated and co-exist within the same somites, with only Mc1 persisting during secondary myogenesis. Lineage tracing using *Mymx:Cre; RosaTdT* mice demonstrated that the Mc2 cells arise from the Mc1. Mechanistically, we show that Mef2 and Rxr factors positively and negatively regulate *Mymx* expression, respectively. Additionally, RXRG interacts with MYOD1 and MYOG, modulating their transcriptional activity in luciferase assays. Collectively, our findings uncover two populations among the myocytes that drive primary and secondary fiber formation, challenging the traditional view of vertebrate muscle precursor homogeneity.

KEY WORDS: Myogenesis, Skeletal muscle, Myomaker, Myomixer, Myocyte, Differentiation dynamics, Mouse

## INTRODUCTION

Skeletal muscle formation during embryogenesis, postnatal maintenance and regeneration after injury relies on muscle precursor cells in embryos, and muscle stem cells, known as satellite cells, in adults (Bentzinger et al., 2012). During each of these processes, precursors first commit to the muscle lineage and generate myoblasts that differentiate into elongated myocytes (Bentzinger et al., 2012). These myocytes then fuse to form multinucleated myotubes, which mature into myofibers, the primary component of skeletal muscle (Petrany and Millay, 2019). The mechanisms underlying myogenic differentiation are relatively well-studied and conserved across invertebrate and vertebrate species (Chal and Pourquié, 2017; Kim et al., 2015). The evolutionarily conserved basic helix-loop-helix (bHLH) family of transcription factors MYF5, MRF4 (also known as MYF6), MYOD1 and MYOG, collectively known as myogenic regulatory factors (MRFs), are sequentially expressed during myogenesis (Mok and Sweetman, 2011). MRFs activate muscle differentiation by binding to the E-box motifs present in the promoter of numerous muscle-specific genes, thereby regulating their expression (Hernández-Hernández et al., 2017). Although some aspects of vertebrate myogenesis have been well-described in the past decades, the early cellular events that initiate embryonic myogenesis remain poorly defined. The prevailing model suggests that muscle precursor cells initially express PAX3, then commit into myoblasts marked by MYF5 and MYOD1 expression, followed by further differentiation into MYOG-expressing myocytes, which ultimately fuse to form syncytial myotubes (Bentzinger et al., 2012). Notably, this model of myogenesis assumes a homogeneous pool of muscle progenitors.

In contrast, the mechanisms of *Drosophila* myogenesis, specifically the first fusion step, has been studied in far greater detail (Richardson et al., 2008; Tixier et al., 2010). During fly embryogenesis, two distinct myoblast populations have been identified: founder cells (FC) and fusion-competent myoblasts (FCM) (Bate, 1990). In the initial phase of myogenesis, neighboring FCMs are attracted to a single FC and fuse with it to form a multinucleated syncytium (Artero et al., 2001; Rau et al., 2001; Bour et al., 2000; Shelton et al., 2009). Following fusion, the nuclei of the fusing FCMs adopt the transcriptional identity of their respective FC (Ciglar et al., 2014; Bothe and Baylies, 2016). Each of the 30 FCs in a given hemi-segment dictates the transcriptional identity, position, orientation and size of the corresponding embryonic body wall muscles (Beckett and Baylies, 2007). The gene expression and functional diversity of *Drosophila* embryonic myoblasts are paralleled by a diversity in their cell surface proteomes. Notably, cell-cell fusion is a complex, multistep process requiring adhesion molecules on either side of the fusion synapse to establish close contact between membranes. The essential adhesion molecules Duf (Kirre) and Rst in the FC and Sns and Hbs in FCMs are well-characterized for their roles in engaging the actin cytoskeleton. These interactions facilitate cytoskeleton rearrangement, generating protrusive forces on one membrane and resistance forces on the other, ultimately leading to membrane mixing, fusion pore formation and complete cell fusion (Kim and Chen, 2019).

The vertebrate orthologues of the *Drosophila* adhesion proteins, Nephrin (Nphs1) and Neph1-3 (Kirrel1-3), do not appear to have a conserved evolutionary function in primary myogenesis. Instead,

[1]Programmes de Biologie Moléculaire, Université de Montréal, Montréal, QC H3T 1J4, Canada. [2]Institut de Recherches Cliniques de Montréal, 110, Avenue des Pins-Ouest, Montréal, QC H2W 1R7, Canada. [3]Department of Basic Neurosciences, University of Geneva, CH-1211 Geneva, Switzerland. [4]Département de Biochimie, Université de Montréal, Montréal, QC H3T 1J4, Canada. [5]Institut NeuroMyoGène, MéLiS, CNRS UMR 5310, INSERM U1217, Université Claude Bernard Lyon 1, 69008 Lyon, France. [6]Biologie du développement et cellules souches, Institut de Génétique et de Biologie Moléculaire et Cellulaire (IGBMC), Inserm U1258; CNRS UMR 7104; Université de Strasbourg, 1 rue L. Fries, Illkirch 67404, France. [7]Department of Anatomy and Cell Biology, McGill University, Montreal, QC H3A 1A3, Canada. [8]Département de Médecine, Université de Montréal, Montréal, QC H3T 1J4, Canada.

*Author for correspondence ( jean-francois.cote@ircm.qc.ca)

S.N., 0000-0001-6201-6014; J.-F.C., 0000-0001-7055-2642

vertebrates have developed myogenic lineage-specific fusogenic proteins. The first identified was myomaker (MYMK), a seven transmembrane domain protein that dimerizes in the membrane (Millay et al., 2013; Long et al., 2023). Subsequently, myomixer (MYMX), a single-pass transmembrane microprotein, was discovered (Bi et al., 2017; Zhang et al., 2017; Quinn et al., 2017). Lipid membrane experiments demonstrated that MYMK and MYMX are involved in distinct steps during fusion, where MYMK is required first for membrane hemi-fusion and MYMX acts later to promote fusion pore formation (Leikina et al., 2018). Functional studies using depletion and overexpression approaches in myoblasts, as well as in fibroblasts and HEK293 cells, demonstrated that MYMK is required on both fusing cells, while MYMX is needed on only one (Bi et al., 2017; Millay et al., 2014; Quinn et al., 2017). This asymmetry parallels the fusion synapse observed in *Drosophila*. However, the physiological relevance of these cell culture findings *in vivo* remains to be fully elucidated.

Despite three decades of research into the cellular, functional, and transcriptional diversity in *Drosophila*, it remains unclear, and actively debated, whether vertebrates exhibit similar diversity and asymmetry in myocyte fusion (Powell and Wright, 2012; Petrany and Millay, 2019). For example, lineage tracing using *nlacZ*-Myf5[Cre] or Myf6[Cre] reporter mice identified both Myf5-dependent and Myf5-independent putative muscle precursor populations (Haldar et al., 2008; Kablar et al., 2003). Another group also suggested the existence of two myogenic lineages by studying the Myf5-mediated regulation of *Myog* expression (Yee and Rigby, 1993). Additionally, the presence of Pax7[+]/Myf5[+] and Pax7[+] lineages has been proposed in both mouse and chick models (Picard and Marcelle, 2013). Collectively, these studies highlight a considerable interest in this subject, but also suggest that our understanding of the diversity of muscle progenitors in vertebrates remains partial. Previous investigations have primarily focused on a limited set of known myogenic genes. A comprehensive, transcriptome-wide analysis of muscle precursor identity is essential to fully elucidate the early events governing vertebrate myogenesis.

In this study, we used an unbiased single-cell RNA-sequencing (scRNA-seq) approach to re-examine the longstanding question of muscle progenitor diversity with unprecedented resolution. We identified a heterogenous population of early muscle progenitors at embryonic day (E) 9.5 differentially expressing *Ifitm1* and *Myf5* and revealed that IFITM1 functions to restrict fusion of unmatured cells. In matured progenitors, we identified two distinct myocyte states: Myocyte 1 (Mc1), expressing both *Mymk* and *Mymx*, and Myocyte 2 (Mc2), expressing only *Mymk*. The co-existence of Mc1 and Mc2 cells within the same somites was confirmed before primary myogenesis, whereas only the Mc1 were maintained during the secondary myogenesis. Additionally, cell lineage tracing analysis revealed that the Mc2 were the progeny of the Mc1. Mechanistically, we investigated the regulation of *Mymx* expression in these populations, we found that MEF2A acts as an activator, whereas RXRG functions as a repressor, likely through interactions with MYOD1 and MYOG. Collectively, these findings reveal previously unrecognized myocyte subpopulations and regulatory mechanisms during embryonic myogenesis, challenging the traditional view of muscle progenitor homogeneity.

## RESULTS

### Single-cell transcriptomic atlas of muscle progenitors in E9.5 and E11.5 mouse embryos

To investigate the longstanding question of cellular diversity within embryonic muscle progenitors in vertebrates, we dissected inter-limb somites from E9.5 and E11.5 mouse embryos, corresponding to stages of muscle lineage commitment and the pre-fusion phase, respectively (Fig. 1A). Dissociated somite cells were subjected to scRNA-seq, generating transcriptomic datasets of 10,000 cells per condition, with an average of 30,000 reads and 3500 detected genes per cell. Datasets from the E9.5 and E11.5 embryos were integrated using R-Seurat and clustered based on differentially expressed genes, and cell clusters were manually annotated using publicly available databases (Fig. 1B). Muscle progenitor clusters were identified by their expression of early muscle progenitor (myoblasts) markers, such as *Pax3*, *Pax7* and *Myf5*, and late muscle progenitor (myocytes) markers, including *Myod1*, *Myog* and *Myh3* (Fig. 1C). Myoblast populations were predominant in E9.5 embryos, whereas myocytes were more abundant at E11.5 (Fig. 1D). We identified a continuum of muscle progenitor populations, including Myoblast 1 and 2 and Myocyte 1 and 2, which align with the pseudotime analysis. This analysis revealed a potential transcriptional trajectory from Myoblast 1 to Myoblast 2, further following to Myocyte 1 and ultimately Myocyte 2 (Fig. 1C,E).

The expression of muscle progenitor markers is dynamic during these developmental stages (Mok and Sweetman, 2011). Our analysis revealed that *Pax3*-expressing myoblasts remained relatively stable, constituting ∼76% of the population at E9.5 and 60% at E11.5. In contrast, *Pax7*-expressing myoblasts showed a significant increase, rising from 8.7% at E9.5 to 43.5% at E11.5 (Fig. 1F; Table S1). Additionally, the percentage of *Myf5*-expressing cells remained relatively stable. Markers of late muscle progenitors, such as *Myod1*, *Myog* and *Myh3*, were mostly expressed at E11.5 (Fig. 1G), indicating the transition to more differentiated muscle cell states. These results demonstrate that we have generated high-quality datasets, providing a valuable resource to investigate the diversity of muscle cell populations during primary myogenesis.

### Transcriptional profiling of E9.5 muscle progenitors identifies four distinct subpopulations

To investigate the diversity among myoblast populations, E9.5 muscle progenitors were extracted and subclustered (Fig. 2A), resulting in the identification of four distinct subclusters, all of which expressed *Pax3* (Fig. 2B). One subcluster was characterized by the expression of developmental genes such as *Hoxa9*, *Dll3*, *Notch1* and *Lef1*, but lacked *Myf5* expression. This subcluster was therefore designated as the muscle precursor population (Fig. 2C,D). The remaining subclusters included two proliferative myoblast populations, termed Cycling myoblasts 1 (S phase) and 2 (M phase), distinguished by the expression of proliferation markers. Cycling myoblasts 1 expressed markers such as *Mcm3* and *Cdc6*, while Cycling myoblast 2 expressed markers like *Cdk1*, *Aurka* and *Mki67* (Fig. 2C). The final subcluster, identified as the myoblast population, expressed both developmental and myogenic genes, including *Wnt6* and *Myf5* (Fig. 2B,C). Consistent with previous findings, we observed a continuum of cell states transitioning from muscle precursors expressing Notch pathway components to committed myoblasts expressing *Myf5* (Bjornson et al., 2012; Delfini et al., 2000; Esteves de Lima et al., 2016; Mourikis et al., 2012).

Interestingly, *Ripply2* and *Cer1* were uniquely expressed in a mutually exclusive manner in the muscle precursor population (Fig. 2C,D). Ripply2 plays a crucial role in somitogenesis, while Cer1, a secreted ligand of the TGFβ superfamily, functions as a bone morphogenic protein antagonist (Chi et al., 2011; Chan et al., 2007). Additionally, expression of components of the Notch signaling pathway, including *Notch1*, *Dll3* and *Lef1*, were significantly enriched in the muscle precursor population (Fig. 2D). The ligand Dlk1 was

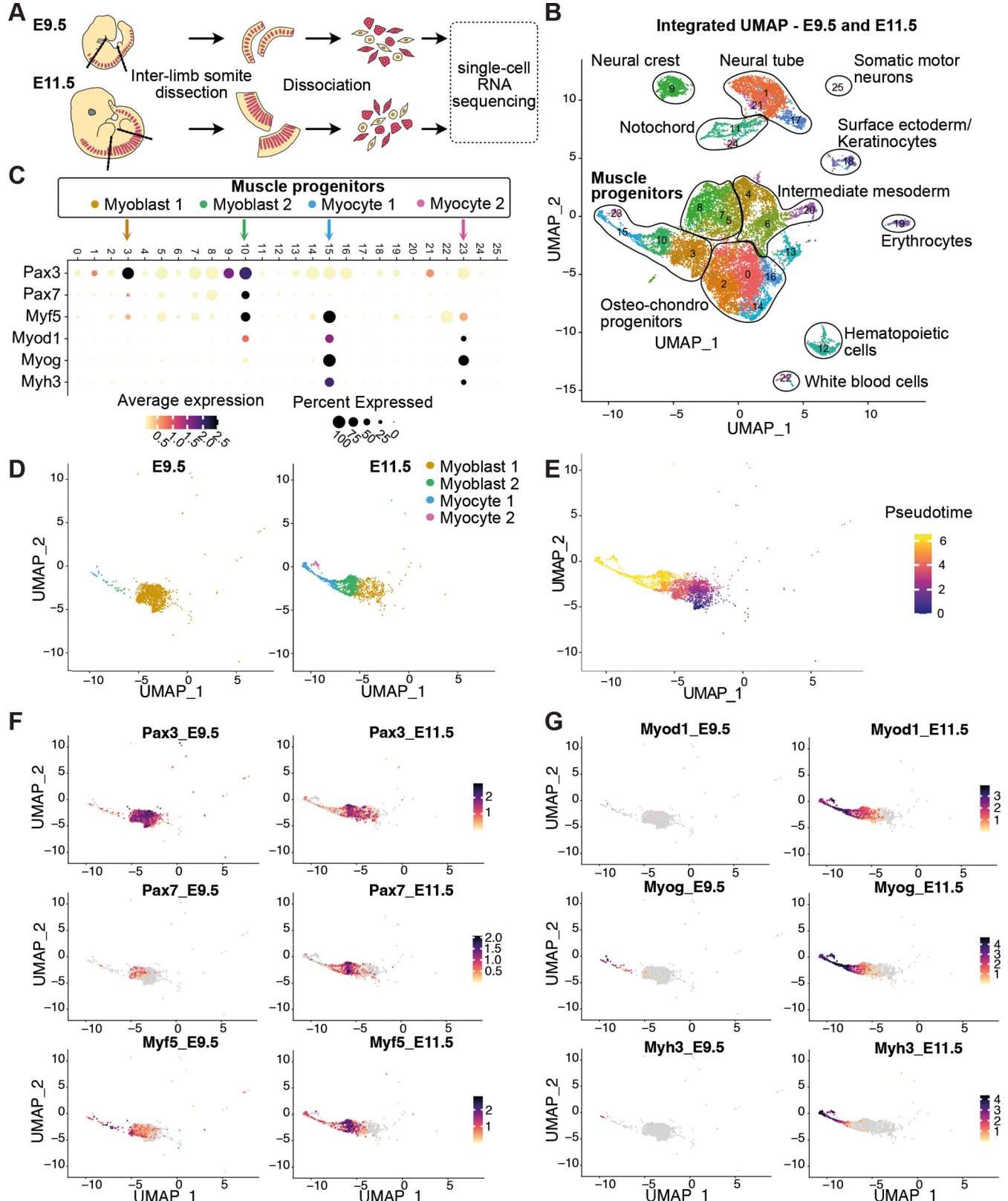

Fig. 1. Single-cell transcriptomic atlas of muscle progenitors in E9.5 and E11.5 mouse embryos. (A) Inter-limb somites from E9.5 and E11.5 mouse embryos were dissected, dissociated and processed for scRNA-seq. (B) Integrated UMAP of E9.5 and E11.5 datasets showing all the somites identified. The central clusters contain cells found within the somites. (C) Dotplot showing the early and late muscle progenitors marker expression in the identified clusters. (D) Split UMAP showing the muscle progenitor subset in E9.5 and E11.5 mouse embryos. The Myoblast 1 cluster is enriched in E9.5 whereas Myoblast 2 and the myocytes are enriched in E11.5. (E) Pseudotime analysis of the muscle progenitor subset using Monocle3. The predicted differentiation order correlates with the differentiation state of muscle progenitor cells (F,G) Split feature plots showing the expression of early (F) and late (G) muscle markers in the muscle progenitor clusters.

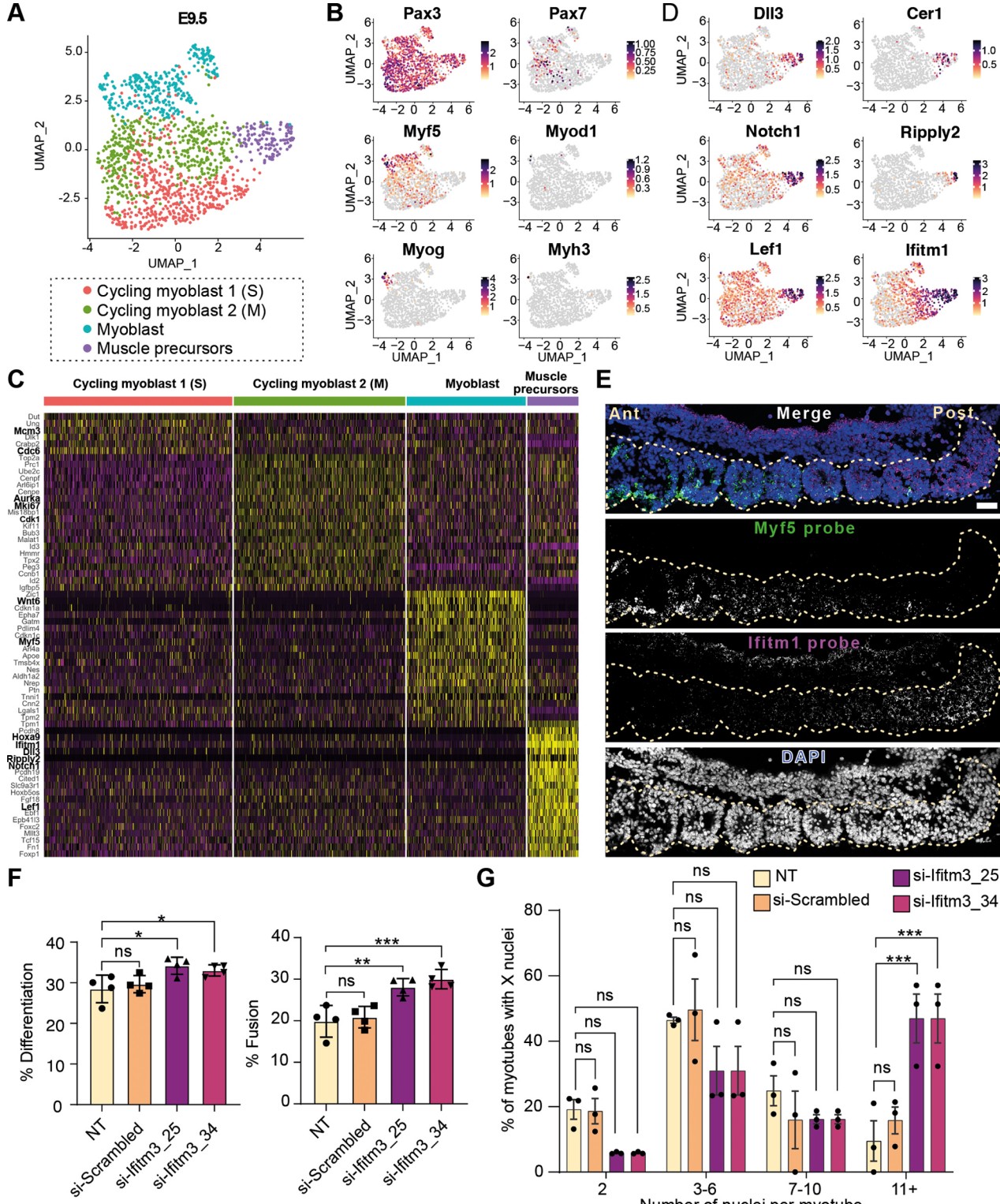

**Fig. 2. Transcriptional profiling of E9.5 reveals that muscle progenitors transiently express the fusion inhibitor Ifitm1 to restrict fusion.** (A) UMAP showing the sub-clustering of the muscle progenitor subset in E9.5 mouse embryos. (B) Feature plots showing the expression of established muscle progenitor markers. (C) Heat maps showing the top differentially expressed genes of each subpopulation. The yellow and magenta colors represent a high and low expression, respectively. (D) Feature plots showing the expression of novel markers of the muscle precursor population. (E) Fluorescence *in situ* hybridization against Ifitm1 (magenta) and Myf5 (green) RNA in E9.5 embryo sections. The shown section contains a sequence of somites from a representative example of *n*=3 embryos. Dashed line indicates the somites spanning the length of the embryo section. Scale bar: 40 μm. (F) Bar graph showing the percentage of differentiation (nuclei in MHC+ cells/total DAPI stained cells) and fusion (MHC+ cells with at least two nuclei/total DAPI stained cells) following the knockdown of *Ifitm3* in C2C12 using siRNA. (G) Quantification of percentage of nucleation in differentiated C2C12. Each data point represents an independent experiment (*n*=4). Two-tailed unpaired *t*-test was performed between the control and the treated condition. Error bar represents the s.d. ns, non-significant. *$P<0.05$, **$P<0.01$, ***$P<0.001$.

exclusively expressed in the myoblast populations, but not the muscle progenitor population. Conversely, all three Notch receptors (*Notch1*, *2* and *3*) were almost exclusively expressed in the muscle precursor population (Fig. S1A). Given that the Notch signaling pathway is known to regulate stem cell maintenance and lateral inhibition to generate distinct cell types, we used CellChat to explore potential population interactions based on ligand-receptor expression (Fig. S1B,C) (Jin et al., 2021). Our analysis revealed significant trans Notch signaling between the cycling myoblast populations (Cycling myoblasts 1 and 2) and the committed myoblast population on one side, and the muscle progenitors on the other (Fig. S1B). In addition, cis Notch signaling was observed within the muscle progenitor population itself. Notably, the most prominent ligand-receptor pairs included Dlk1-Notch2, Dlk1-Notch1 and Dll3-Notch2 (Fig. S1C). The roles of these highlighted genes, particularly the newly identified Notch interactions, in regulating primary myogenesis remain largely unexplored, presenting a promising area for future research.

### Muscle precursors transiently express the fusion inhibitor Ifitm1 to restrict fusion

The gene coding for the interferon-induced transmembrane protein IFITM1 was notably highly expressed in the muscle precursor population, and its expression gradually decreased as *Myf5* expression increased (Fig. 2D). To localize *Ifitm1*- and *Myf5*-expressing cells within E9.5 somites, we performed fluorescence *in situ* hybridization. Longitudinal sections of E9.5 embryos allowed us to construct a developmental timeline, as anterior somites segregate and differentiate earlier than posterior ones. Our analysis revealed low *Ifitm1* expression in the anterior somites, with progressively higher expression in the posterior somites. In contrast, *Myf5* exhibited the opposite pattern, with high expression in anterior somites and a gradual decline posteriorly (Fig. 2E; quantified in Fig. S2A). These observations corroborate the scRNA-seq data and provide additional spatial resolution for *Ifitm1* and *Myf5* expression.

Given that IFITM proteins are known to inhibit fusion in various contexts, including trophoblasts fusion and virus fusion for entry in host cell (Zani et al., 2019; Degrelle et al., 2023; Gómez-Herranz et al., 2023), we hypothesized that *Ifitm1* downregulation, coinciding with *Myf5* expression, might act as a mechanism to regulate the timing of myoblast fusion. In adult satellite cells, we found low expression levels of *Ifitm2* and high levels of *Ifitm3* upon activation in injured muscles, with these levels decreasing as differentiation progressed (Fig. S2B) (McKellar et al., 2021). However, *Ifitm1* levels were undetectable in adult muscle (de Micheli et al., 2020). The expression of IFITM3 protein was confirmed in a C2C12 myoblast cell line 24 h after induction of differentiation (Fig. S2C). To test our hypothesis, we depleted *Ifitm3* in C2C12 using two different small interfering RNAs (siRNA) and quantified differentiation and fusion (Fig. 2F,G and Fig. S2D,E). Knockdown of *Ifitm3* slightly increased the percentage of differentiated myoblasts and significantly enhanced fusion. Notably, the proportion of myotubes containing at least 11 nuclei was markedly increased (3-fold) in *Ifitm3*-depleted conditions, compared to controls, confirming the role of IFITM proteins in myoblast fusion (Fig. 2F,G). Collectively, these results identify a new continuum between muscle progenitors and myoblasts, characterized by an inverse expression of *Myf5* and *Ifitm1*. This dynamic likely plays a role in coordinating the timing of myoblast fusion during embryonic myogenesis.

### Myomixer expression is lacking in a subpopulation of myocytes at E11.5

To investigate cellular diversity among embryonic myocytes, we further subclustered the E11.5 muscle progenitor populations

(Table S2). We identified seven distinct subclusters (Fig. 3A). Four of these were classified as myoblasts based on the expression of *Pax3*, *Pax7* and *Myf5*, while two were identified as myocytes based on the expression of *Myod1*, *Myog* and *Myh3* (Fig. 3B). The seventh cluster was excluded as it lacked specific markers and exhibited low total RNA and gene counts, an evident technical artifact of no biological significance (Fig. S3A). The myoblast populations were subdivided based on their expression profiles of specific developmental genes. Myoblast 1 expressed *Pax3* and *Foxp1*, while Myoblast 2 expressed *Prrx1* and *Twist1* (Fig. 3C). Additionally, two cycling myoblast populations were distinguished by their cell cycle gene expression. Cycling myoblast 1 expressed *Mcm6* and *Ccnd2*, markers associated with the G phase, while Cycling myoblast 2 expressed *Cdk1* and *Cdc20*, associated with the M phase of the cell cycle (Fig. 3C) (Vermeulen et al., 2003; Loddo et al., 2009).

Remarkably, two distinct myocyte populations emerged from this analysis, which we designated as Mc1 and Mc2. Mc1 expressed genes related to cellular growth arrest and early differentiation, such as *Gadd45g*, *Cdkn1a* and *Myod1*, while Mc2 expressed genes associated with muscle cell contraction and terminal differentiation, including *Acta1*, *Trdn*, *Myh7*, *Myh3* and *Ldb3* (Fig. 3D and Fig. S3B,C). To further characterize Mc1 and Mc2 myocyte populations, we performed gene ontology (GO) analysis on the top 100 differentially expressed genes (DEGs). For Mc1, the top biological process terms included 'myoblast differentiation' and 'muscle tissue development' (Fig. S3D). In contrast, 'muscle contraction' was the predominant term for the Mc2, appearing in six GO classifications. Analyzing the top molecular function of the Mc1 population, we found genes primarily associated with transcription regulation, including terms like 'DNA-binding transcription factor binding' and 'E-box binding' (Fig. S3E). In contrast, Mc2 displayed enrichment in structural protein-related terms, such as 'Actin binding', 'Titin binding', 'Actinin binding', 'Telethonin binding' and 'Tropomyosin binding', all essential for sarcomeric Z-line organization and muscle contraction (Fig. S3E). Closer examination of the DEGs heatmap revealed that Mc1 markers were not completely downregulated in Mc2, and vice versa, suggesting a potential transition between the two populations (Fig. 3D). This observation was further supported by the differentiation trajectory prediction generated using Monocle 3, which suggests a linear trajectory where Mc1 differentiate into Mc2 (Fig. 3E). Together, these results suggest that Mc1 represents a transitional state in muscle differentiation, while Mc2 corresponds to a more terminally differentiated, postmitotic, contractile myocyte state.

Upon differentiation, myocytes fuse to form multinucleated muscle fibers. Recent studies identified Mymk and Mymx as muscle-specific fusogens that are essential for cell-cell fusion (Millay et al., 2013; Bi et al., 2017; Zhang et al., 2017; Quinn et al., 2017). Interestingly, we observed that *Mymk* was expressed in both myocyte populations, whereas *Mymx* expression was restricted to Mc1 (Fig. 3D,F and Fig. S4A,B). This observation strikingly aligns with findings from *in vitro* cell-cell fusion assays using myoblasts and heterologous cell systems, which showed that Mymk is required on both fusing cells, whereas Mymx is needed on only one (Quinn et al., 2017; Bi et al., 2017; Millay et al., 2014). To map the localizations of the Mc1 and Mc2 populations in the developing embryo, we performed fluorescence *in situ* hybridization. As somite maturation progresses from the anterior to the posterior end of the embryo, we hypothesized that population transitions would reflect shifts in the Mc1/Mc2 ratio across somites. While the close proximity of fibers precluded precise quantification, we observed $Mymk^+;Mymx^+$ and $Mymk^+;Mymx^-$ myocytes, consistent with our

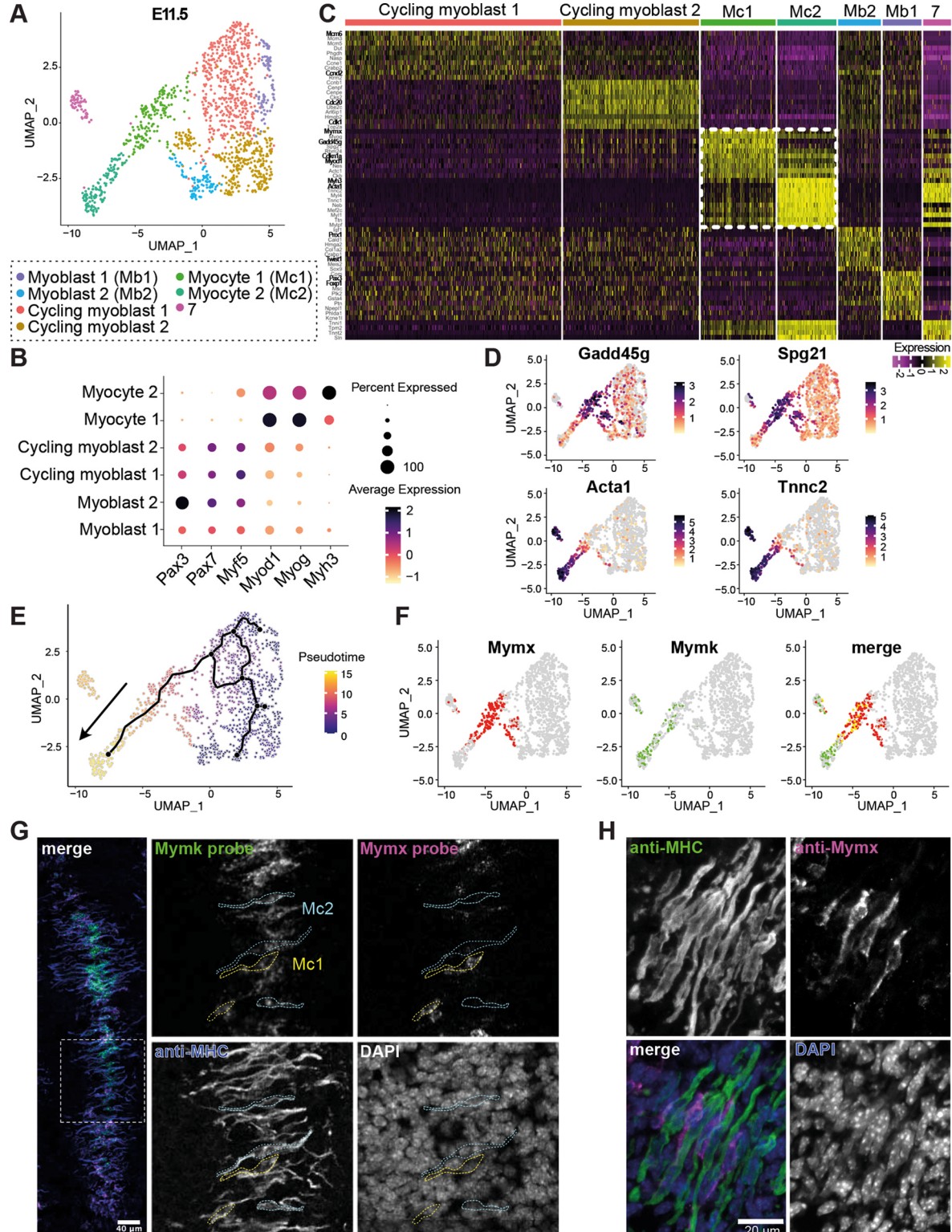

**Fig. 3. Myomixer expression defines distinct myocyte subpopulations at E11.5.** (A) UMAP showing the sub-clustering of the muscle progenitor subset in E11.5 mouse embryos. (B) Dot plot showing the expression of early and late muscle marker genes. (C) Heat map showing the top differentially expressed genes (DEGs) in each muscle subpopulation. Each vertical color represents a cell. The yellow and magenta colors represent a high and low expression, respectively. Subpopulation 7 represents an artifact population and has no exclusive markers. Boxed area indicates Mc1 and Mc2 markers. (D) Feature plot showing the expression of proliferation-related genes in Mc1 and muscle contraction-related proteins in Mc2. (E) Pseudotime superposed with prediction of the differentiation trajectory between the early and the late muscle progenitors using Monocle3. (F) Feature map showing the binary expression of Mymx and Mymk and the merge between the two. Mymx is expressed in the Mc1 while Mymk is expressed in both the Mc1 and Mc2. (G) Fluorescence *in situ* hybridization against Mymk (green) and Mymx (magenta) in a E11.5 somite section co-stained with the MF20 antibody (recognises all myosins heavy chain, MHC). Mc1 and Mc2 cells are highlighted in yellow and blue dotted lines respectively. (H) Immunofluorescence against MYMX in magenta and MHC in green.

scRNA-seq findings, supporting the co-existence of Mc1 and Mc2 populations within the same somite (Fig. 3G and Fig. S4C). To further validate muscle progenitor cell state diversity at the protein level, we performed immunofluorescence staining on E11.5 somites. Mymx, a marker for Mc1, and myosin heavy chain (MHC), a marker for Mc2, revealed Mymx⁺;MHC⁻ cells (Mc1) interspersed with Mymx⁻; MHC⁺ cells (Mc2) (Fig. 3H). *Ex vivo* cultures of dissociated somite cells showed that most MHC⁺ and MHC⁻ cells were mononucleated (Fig. S4D,E), suggesting that *Mymx* downregulation is not solely a consequence of fusion. Notably, Mymx⁺ cells exhibited low levels of MHC expression, confirming that they are not terminally differentiated (Fig. S4D). Additionally, to predict if and how those cells might interact together, we performed a CellChat analysis revealing a high probability of interaction between Mc1 and Mc2 populations through the NCAM, JAM and Cadherin signaling pathways, suggesting that these cell populations likely physically interact (Fig. S5A, differentiation trajectory). These results suggest that the initial Mc1 state transitions to the subsequent Mc2 state. During this differentiation process, we observed downregulation of genes such as *Mymx*, *CD82*, *Mef2a*, *Myod1* and *Myog*, and an upregulation in *Myh3*, *Cd36*, *Mef2c* and *Rxrg* (Table S2). Notably, both cell states co-exist within developing mouse somites and are spatially positioned near one another.

### Persistence and self-renewal of the Mc1 myocyte population during secondary embryonic myogenesis

To assess whether the Mc1 and Mc2 populations persist during the secondary myogenesis taking place between E14.5 and E18.5, we first explored their presence in E14.5 mouse embryos. Immunofluorescence staining of E14.5 longitudinal limb sections revealed that Mc1 (Mymx⁺) are predominantly localized in the central region of developing muscles, with minimal presence in the distal extremities (Fig. 4A and Fig. S6A,B). Transversal limb sections showed Mc1 positioned near MHC⁺ myofibers (Fig. 4B). Next, we used a combination of fluorescent RNA probes, specific to *Mymx* and *Mymk*, and an antibody against MHC to investigate the Mc2 population (Mymx⁻; Mymk⁺) at E14.5. We identified Mc1 (Mymx⁺; Mymk⁺) positioned on the MHC⁺ myofibers (Fig. 4C). Notably, all unfused cells were Mc1, not Mc2. The persistence of Mc1 cells at E14.5 prompted us to investigate whether this population is sustained through proliferation. Immunostaining for the proliferation marker Ki67 at multiple developmental stages revealed that 40-50% of the Mc1 are Ki67⁺ in E11.5, E14.5 and E18.5 mouse embryos (Fig. 4D,E and Fig. S6C,D). Indeed, it was recently demonstrated that, contrary to the established paradigm, MYOG⁺ myocytes can divide, albeit at a limited frequency (Benavente-Diaz et al., 2021). These findings indicate that approximately half of the Mc1 population remains proliferative during both primary (E11.5) and secondary (E14.5, E18.5) myogenesis, supporting the replenishment and maintenance of this population throughout embryonic myogenesis. These results suggest that the non-proliferative fraction of the Mc1 population exit the cell cycle and integrate the growing myofibers through fusion, concomitantly repressing *Mymx* expression.

### The Mc2 are the progeny of the Mc1

To provide direct evidence of the transition from Mc1 to Mc2 and rule out the possibility of an independent lineage branch for each population, we generated a knock-in mouse line expressing Cre under the control of the *Mymx* regulatory elements (Fig. 5A). In these mice, both copies of the *Mymx* gene were preserved, and a T2A-Cre sequence was inserted at the 3′ end of the gene using CRISPR-Cas9. The *Mymx*-Cre mice were crossed with Rosa26^nls-tdTomato reporter

mice to generate embryos expressing a nuclear fluorescent tdTomato protein under the control of the *Mymx* regulatory elements, enabling lineage tracing of the Mc1 cells and their progeny (Fig. 5B).

We verified the integration of the T2A-Cre sequence by PCR amplification of the 5′ and 3′ junctions, followed by sequencing to confirm the absence of mutations in both founder and F1 mice (Fig. 5C). Subsequent progeny was genotyped using conventional Cre primers (Fig. 5D). Cre expression was assessed by immunofluorescence in E11.5 mouse embryos, revealing it is exclusively expressed in Mymx⁺ Mc1 within somites (Fig. 5E). To trace the progeny of Mc1 cells, we performed immunofluorescence staining in E11.5 embryos for Mymx and MHC as markers of Mc1 and Mc2, respectively (Fig. 5F). In Mc1 expressing Mymx, no detectable NLS-tdTomato expression was observed, suggesting that either Cre-mediated recombination of the STOP signal before the *nls-tdTomato* gene had not yet occurred or that the fluorescent protein levels were below the limit of detection. Indeed, previous studies demonstrated the existence of a 0.5- to 0.75-day delay between the start of Cre expression and the onset of recombination (Scotti et al., 2015). In contrast, MHC⁺ Mc2 cells contained nuclei expressing NLS-tdTomato, indicating that Mc2 are derived from Mc1 progenitors (Fig. 5F). To validate NLS-tdTomato expression in Mc1, we looked at E14.5 limb sections and found that both the Mc1 cells and the myofibers express NLS-tdTomato (Fig. 5G). These results confirm our hypothesis that Mc2 arise from Mc1, supporting a linear differentiation trajectory between these two populations.

We propose a model in which Mc1 and Mc2 cells co-exist within the same somites during primary myogenesis (E11.5). In this model, a pool of Mc1 either self-renews or differentiate into Mc2. During secondary myogenesis (E14.5), only the Mc1 population persists, while Mc2 are absent, as they have formed the muscle fibers. Based on their spatial proximity to myofibers, we propose that Mc1 further differentiate and fuse into myofibers. Although Mc1 express both Mymk and Mymx, which should be permissive to fusion, it remains unclear whether Mc1 fuse together. Additional *in vivo* and *ex vivo* experiments will be necessary to explore this possibility.

### Mef2c and Rxrg gene regulatory networks drive the Mc1 to Mc2 transition

To investigate the regulatory mechanisms underlying the transition between muscle progenitor cell states, we performed a regulon analysis (pySCENIC) using the E11.5 scRNA-seq dataset. Cells were clustered based on differentially activated regulons, and the top activated regulons were ranked according to their specificity scores (Fig. 6A,B). Focusing on the transition from Mc1 to Mc2, we examined clusters relevant to these populations. In the Mc1 population, the top three active regulons were Sox8, Snai1 and Lef1, while in the Mc2 population, Rxrg, Hox10 and Mef2c were most prominent (Fig. 6B,C). To further explore this transition, we used CellRouter to reconstruct the cell-state transition trajectory from Mc1 to Mc2 (Lummertz da Rocha et al., 2018; Lummertz da Rocha and Malleshaiah, 2019). Key regulators identified in the transition included transcription factors previously associated with myogenesis, such as Arx, Rxrg and Mef2c, which were predicted to be important for the Mc2 state, while Myod1, Mef2a and Myog are significant for the Mc1 state (Fig. 6D). Notably, Rxrg and Mef2 emerged as prominent regulators in both the regulon and the CellRouter analyses, suggesting they may be involved in the Mc1 to Mc2 transition. Consistently, expression analysis revealed that *Rxrg* and *Mef2c* expression correlated with *Mymk* expression, while *Mef2a* correlated with *Mymx* expression (Fig. 6E).

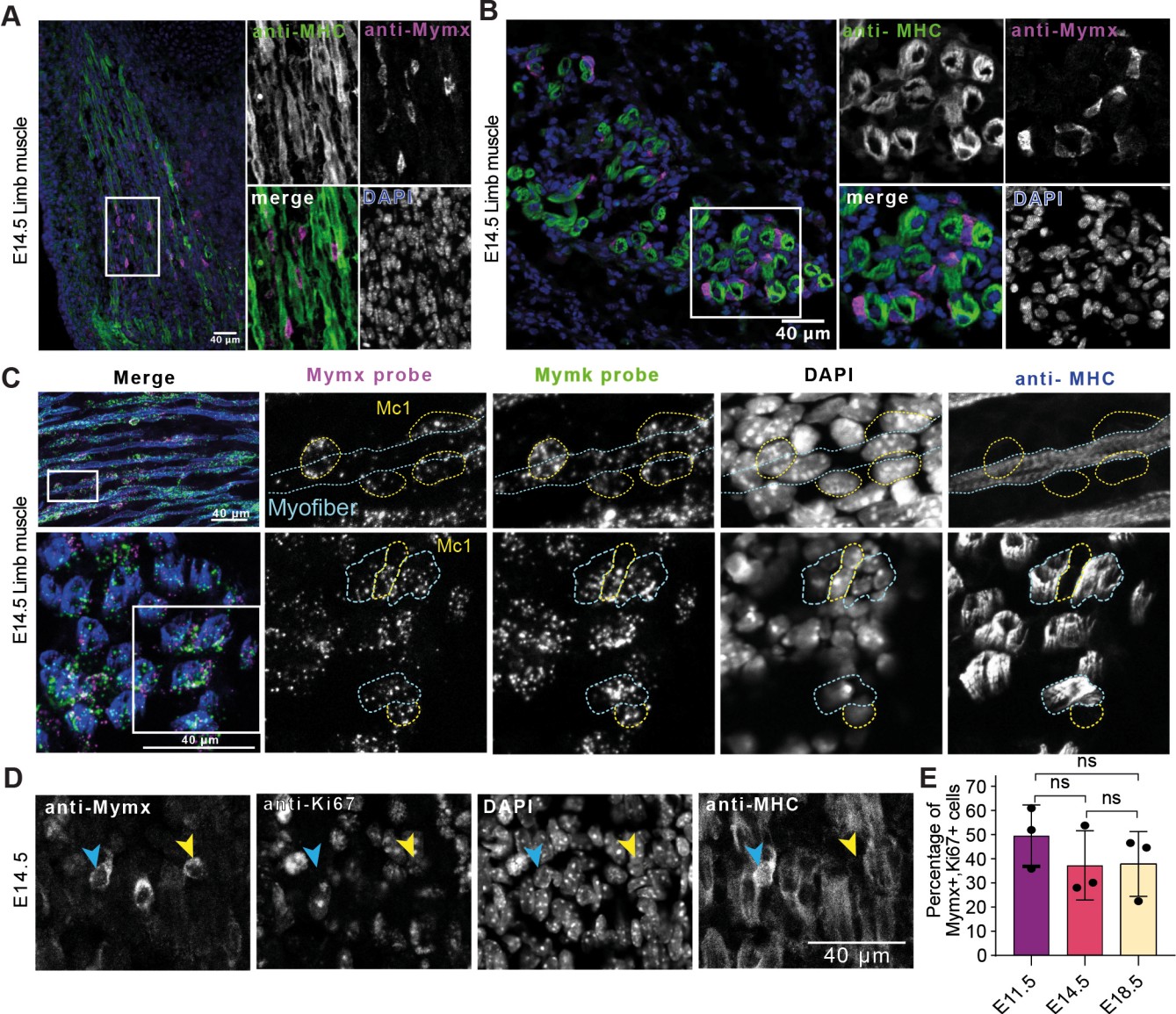

**Fig. 4. Persistence and self-renewal of the Mc1 myocyte population during secondary embryonic myogenesis.** (A,B) Immunofluorescence against MYMX (Magenta) and MHC (green) in E14.5 longitudinal (A) and transversal (left) (B) limb sections. MYMX⁺ cells are localized in between the myofibers. Longitudinal section shows MYMX⁺ cells are situated preferably in the midsection of muscles. (C) Fluorescence *in situ* hybridization against Mymk (green) and Mymx (magenta) in E14.5 limb sections Top and bottom images represent longitudinal and transversal sections, respectively. Sections were also stained with the MF20 antibody against MHC. The Myofiber is highlighted with blue dotted lines and the Mc1 with yellow dotted lines. (D) Immunofluorescence in E14.5 limb sections against Mymx, Ki67 and MHC. Blue and yellow arrowheads indicate MYMX⁺/Ki67⁻ and Mymx⁺/Ki67⁺ cells, respectively. (E) Quantification of the percentage of MYMX⁺/Ki67⁺ cells in E11.5 somites and E14.5 and E18.5 limb sections. One-way ANOVA was performed. Data represent the mean±s.d. *n*=3 embryos per condition. Each point on the graph represents the percentage of Ki67⁺ cells among 150-200 MYMX⁺ cells per embryo. ns, non-significant.

Next, we investigated the mechanisms underlying the transition from Mc1 to Mc2 during myogenesis. A key event in this transition is the induction of *Mymx* expression in Mc1, followed by its downregulation in the Mc2 population, as terminal differentiation begins. To identify positive regulators of Mymx expression, we first examined transcription factors activated in Mc1. Of those, *Mef2a* was particularly interesting as its expression dynamics closely correlated with *Mymx* expression (Fig. 6E). To assess whether *Mef2a* plays a role in regulating *Mymx* expression, we employed differentiating C2C12 as a model for myogenesis. *Mef2a* or *Mef2c* expression was knocked down using siRNAs. The knockdowns reduced the expression of *Mef2a* mRNA by 40% and of *Mef2c* mRNA by 82% (Fig. S7A). As predicted, *Mef2a* knockdown

significantly decreased *Mymx* expression by 35%, without affecting *Mymk* expression, showing that *Mef2a* is required for the normal expression of *Mymx* (Fig. 6F). These findings suggest that a gene regulatory network (GRN) governed by Mef2a contributes to the proper expression of *Mymx* in the Mc1 cell state.

Next, we sought to identify how *Mymx* expression is downregulated during the Mc1 to Mc2 transition. Building on our findings that Mef2a promotes *Mymx* expression in Mc1, we initially tested whether Mef2c could regulate *Mymk* expression. However, knockdown of *Mef2c* did not alter the expression of either *Mymk* or *Mymx* (Fig. 6F). We then focused on *Rxrg*, a top regulon in the Mc2 state with exclusive expression in this population (Fig. 6E). Recently, MYOG and MYOD1 were reported to bind E-boxes in

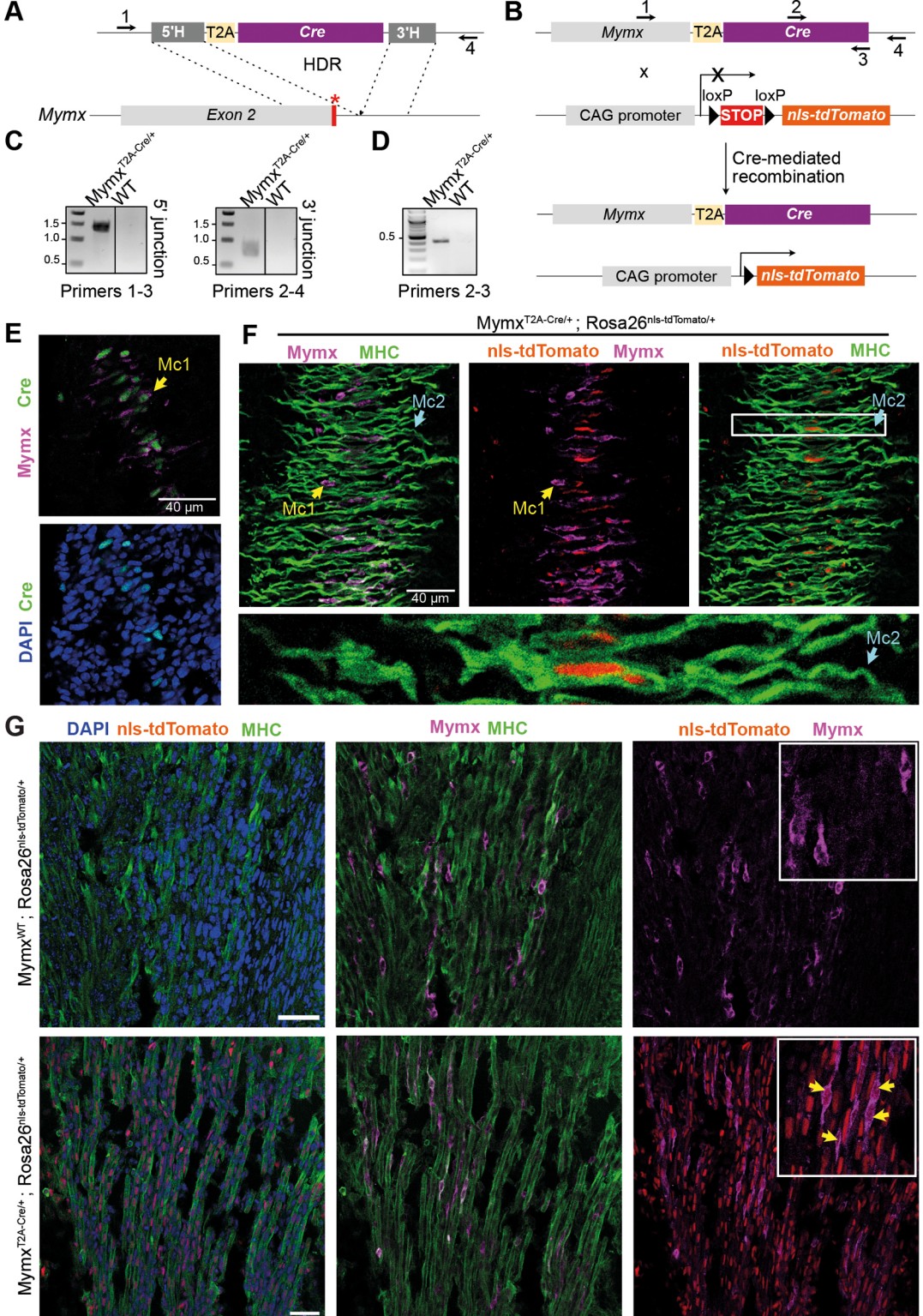

**Fig. 5. The Mc2 are the progeny of the Mc1.** (A) Schematic showing the knocked-in sequence and the homology-directed repair, guided by the 3′ and 5′ homology arms (HA), at the last exon of *Mymx*. (B) Schematic showing the mice crossing to generate cells expressing NLS-tdTomato upon expression of the Cre which is itself under the control of the *Mymx* regulatory region. Numbered arrows indicate the primers used for PCR. (C) PCRs with primers 1-3 and 2-4 to validate the 5′ and 3′ HA and insertion of the T2A-Cre in the right place. The PCRs were purified and sequenced to rule out mutations. (D) PCR used for genotyping using primers 2 and 3. Those primers amplify part of the Cre sequence. (E) Immunofluorescence showing the expression of the Cre (anti-Cre antibody, green) and of MYMX (anti-MYMX antibody, magenta) in a E11.5 embryo section at the somite level. The Cre expression is specific to MYMX+ cells. (F) Immunofluorescence showing lineage tracing of Mc1 in a E11.5 (MymxT2A-Cre+/− ; Rosa26nls-tdTomato/+) section at the somite level. Mc1 are stained with anti-MYMX and Mc2 are stained with MF20. The fluorescent protein nls-tdTomato is present in Mc2 nuclei but not yet expressed in Mc1 nuclei. (G) Immunofluorescence showing lineage tracing of Mc1 in a E14.5 longitudinal limb sections. Mc1 are stained with anti-MYMX and myofibers with MF20. The fluorescent protein NLS-tdTomato is present in Mc1 and myofiber nuclei. Yellow arrows (E-G) indicate MC1 cells. Scale bars: 40 µm.

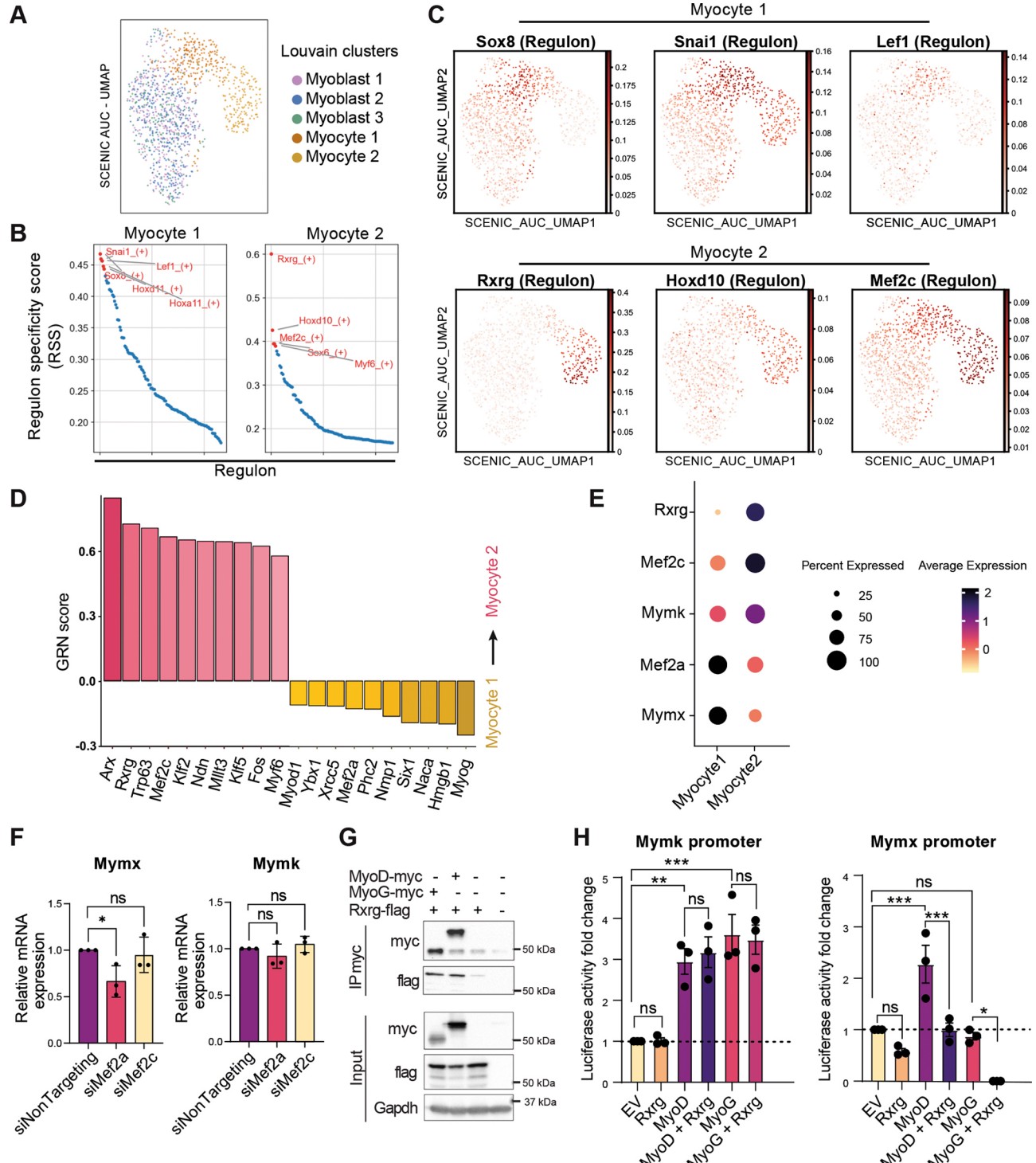

**Fig. 6. Mef2c and Rxrg gene regulatory networks drive the Mc1 to Mc2 transition.** (A) SCENIC AUC-UMAP of E11.5 muscle progenitors generated using pySCENIC for the regulon analysis. (B) RSS panel plot showing the top activated regulons (in red) in the Mc1 and Mc2 populations. (C) Feature map showing the top three activated regulons in the Mc1 (Sox8, Snai1, Lef1) and in the Mc2 (Rxrg, Hoxd10 and Mef2c). (D) CellRouter analysis showing the top transcription factors required for each muscle progenitor state during the Mc1 to Mc2 transition. (E) Dot plot showing the expression of *Rxrg*, *Mef2c* and *Mef2a* relative to the fusogen expression in the myocyte populations. (F) Mymx RNA expression decreased upon knockdown of *Mef2a*. qPCR analysis of Mymx and Mymk expression levels in C2C12 transfected with siRNA targeting Mef2a and Mef2c. $n$=3 independent experiments. The $2^{-\Delta\Delta CT}$ method was used to calculate the relative fold change which was normalized against β-actin expression. Two-tailed unpaired $t$-test was performed between the control and the treated condition. Data represent mean±s.d. ns, non-significant. *$P$<0.05. (G) Co-immunoprecipitation between overexpressed MYOD-myc, MYOG-myc and Rxrg-flag in HEK293T. Rxrg-flag co-immunoprecipitated with MYOD-myc or MYOG-myc. (H) Luciferase assay performed in C2C12. Regulatory regions underlined and highlighted in Fig. S7B were cloned before a luciferase sequence. C2C12 cells were co-transfected with the luciferase vectors along with an empty vector (EV), or vectors expressing Rxrg, Myod or Myog. Luciferase activity fold change is relative to the EV condition. Each dot represents an independent experiment ($n$=3). For the statistics, the ordinary one-way ANOVA-multiple comparisons was performed between the control and the treated conditions. Data represent mean±s.d. ns, non-significant. *$P$<0.05, **$P$<0.01, ***$P$<0.001.

the promoter of *Mymk* to regulate its transcription in zebrafish, chick and human myoblasts (Zhang et al., 2020; Ganassi et al., 2018; Luo et al., 2015). Additionally, MYOD1 was reported to directly bind E-boxes in the promoter of *Mymx* and regulate its transcription in human myotubes (Zhang et al., 2020). Based on these findings, we hypothesized that RXRG might regulate *Mymk* and/or *Mymx* expression by interacting with MYOD1 and/or MYOG transcription factors. To test this, we performed co-immunoprecipitation experiments on overexpressed proteins in HEK293T cells. We found that immunoprecipitation of flag-tagged RXRG robustly co-immunoprecipitated Myc-tagged MYOD1 or MYOG (Fig. 6G). Next, we examined publicly available chromatin immunoprecipitation (ChIP) datasets for RXRG peaks. The only available mouse RXR ChIP data was for RXRA in kidney cells (Meyer et al., 2007). In the *Mymk* promoter region, no prominent RXRA peak overlapped the MYOD1 or MYOG peaks (Fig. S7B). However, a prominent RXRA peak was identified in the *Mymx* 5′-untranslated region sequence, ~200 base pairs from the double E-box situated under the MYOD1 or MYOG peaks. This RXRA peak also contained a retinoic x response element (RXRE), suggesting that RXRG might regulate *Mymx* expression. To test this, we cloned the proximal promoter regions of *Mymx* and *Mymk* genes upstream of a luciferase reporter gene and performed luciferase assays in undifferentiated C2C12. Consistent with previous studies, expression of either MYOD1 or MYOG activated the *Mymk* promoter and induced luciferase expression, whereas only MYOD1, but not MYOG, was able to activate the *Mymx* promoter (Fig. 6H). Rxrg expression alone did not affect the *Mymk* or *Mymx* promoter activity, and co-expression of RXRG with either MYOD1 or MYOG had no effect on the *Mymk* promoter activity, consistent with *Mymk* being expressed in both Mc1 and Mc2 populations. In contrast, co-expression of RXRG with MYOD1 abolished the effect of MYOD1 on the induction of *Mymx* promoter activity (Fig. 6H). While MYOG alone had no effect on the activity of the *Mymx* promoter, co-expression with RXRG completely blocked its basal transcription. Furthermore, the addition of 9-cis retinoic acid, a ligand for RXRG, did not affect the transcriptional regulation through the *Mymx* promoter (data not shown). Collectively, these results demonstrate that RXRG expression contributes to the downregulation of *Mymx* expression, providing a partial explanation for the regulatory mechanisms underlying the Mc1 to Mc2 transition.

## DISCUSSION

The initial cellular mechanisms responsible for skeletal muscle formation in vertebrates remain incompletely understood. Here, we uncover transcriptionally distinct myoblast and myocyte cell states in the developing mouse embryo, redefining the landscape of early myogenesis.

We show that the differentiation trajectory of early muscle progenitors follows a linear progression, with cells initially expressing *Ifitm1* committing to the myoblast lineage by downregulating *Ifitm1* and simultaneously upregulating *Myf5*. These muscle precursors also express components of the Notch signaling pathway, such as *Dll3*, *Notch1* and *Lef1*. The Notch pathway has been demonstrated to act as an inhibitor of muscle differentiation in various models and to repress the expression of cell cycle exit genes such as p21 and p57, therefore promoting the proliferation of progenitor cells (Bjornson et al., 2012; Delfini et al., 2000; Esteves de Lima et al., 2016; Mourikis et al., 2012; Zalc et al., 2014). Our data consolidate the previous finding that Notch signaling is active in undifferentiated muscle cells. A recent study identified a specific role for the Notch effector *Heyl* (hairy and enhancer-of-split related with YRPW motif) in the direct transcriptional repression of *Mymk* in chicken embryos

(Esteves de Lima et al., 2022). This repression likely serves to prevent the fusion of incompletely differentiated cells. Similarly, we found that early myoblast precursors express *Ifitm1*, encoding a transmembrane protein that negatively regulates membrane fusion in a virus-cell and cell-cell fusion context (Prelli Bozzo et al., 2021; Xie et al., 2023). For example, in placental development, trophoblast fusion was blocked by expression of the Ifitm proteins, whereas their knockdown promoted fusion (Degrelle et al., 2023; Zani et al., 2019). Building on these findings and our scRNA-seq dataset, we propose a model in which Ifitm1 expression in poorly differentiated muscle precursors acts as a fusion inhibitor, ensuring the precise timing of myofiber formation. This reveals an additional layer of regulatory control over the cell-cell fusion process, governed by the temporal regulation of Ifitm and *Myf5* gene expression. Supporting this model, we demonstrated that Ifitm3 negatively regulates fusion in C2C12 myoblasts.

Our findings provide compelling evidence that challenges the assumption of homogeneity among vertebrate muscle precursors. By identifying transcriptionally distinct cell states within the myocyte population, we reveal a previously unrecognized layer of complexity in early muscle development in vertebrates. Specifically, we demonstrate that the fusogen Mymx is differentially expressed between two co-existing myocyte populations, Mc1 and Mc2, within the somites at E11.5. This transcriptional divergence underscores a previously unappreciated level of complexity in myocyte identity and function during early muscle development.

By integrating predicted differentiation trajectories with transcriptional profiles, we demonstrate that the Mc1 cell population gives rise to Mc2, establishing a hierarchical relationship between these cell populations. Gene expression analysis further delineates their functional roles, with Mc1 characterized by genes associated with muscle differentiation, while Mc2 cells predominantly express genes involved in muscle contraction and structural assembly. This previously unrecognized stratification suggests that muscle progenitors are not a homogeneous population, but instead possess distinct developmental specializations, reinforcing the concept of functional diversity within somites.

The discovery that *Mymx* is downregulated in the terminally differentiated myocytes is a significant observation, given its essential role in myocyte fusion. This finding challenges the conventional view of a homogeneous fusion process and instead supports an asymmetric fusion mechanism, wherein both Mc1 and Mc2 populations are required for the initial fusion. Notably, our finding that both Mc1 and Mc2 express *Mymk*, while only Mc1 express *Mymx* mirrors *in vitro* fusion assays showing that Mymk is required in both fusing cells, whereas Mymx is needed only in one (Millay et al., 2014; Quinn et al., 2017). Our results provide the first *in vivo* evidence for a fusogen-based myocyte state diversity, showing that primary multinucleated myofibers are generated through Mc1-Mc2 fusion.

Notably, since Mymk is essential for hemi-fusion and Mymx is required for pore formation, we propose a model where only Mc1 can open the fusion pore towards Mc2. In contrast, Mc2 is only capable to be on the receiving end of fusion, as it lacks the pore formation potential but retains the ability to fuse (Leikina et al., 2018). This fusion mechanism parallels the first fusion steps observed in *Drosophila* myogenesis where the FCM attacks the FC and not the reverse.

We further demonstrate that only Mc1 cells persist at later stages of myogenesis, where they preferentially localize to the central region of the developing muscle. This finding aligns with observations in chick embryos, where Mymk-expressing mononucleated progenitors

predominantly localize at the center of muscle fibers (Esteves de Lima et al., 2022), suggesting a model in which fusion-competent myocytes preferentially fuse in the center of muscle fibers. In contrast, earlier studies suggested that myocyte fusion occurs at the muscle tips; however, these studies did not characterize the identity of the fusing cells. (Zhang and McLennan, 1995). More recent studies have clarified that cells with a dual fibroblast-myoblast identity are responsible for fusion occurring at the muscle tips, specifically contributing to the formation of the myotendinous junction (Esteves de Lima et al., 2021; Yaseen et al., 2021).

Using cell lineage tracing, we further demonstrated that Mc1 cells give rise to Mc2, thereby establishing the mechanisms that directly precede the first myoblast fusion events. This finding suggests the possibility that Mc1 terminally differentiate to form the Mc2 population, creating a building block that serves as the scaffold for subsequent fusion events with Mc1. In this model, Mc2 can be conceptualized as 'pioneer myocytes'. In support of this idea, Lee et al. (2013) proposed that 'pioneer myoblasts' differentiate into myocytes and inhibit the differentiation of surrounding cells in amniote primary myogenesis. This mechanism, revealed through a series of Pax3, Pax7, Myog and myosins immunostaining, is

thought to limit the number of primary myocytes formed, ensuring the controlled development of primary muscle fibers. Our finding expands on this concept by implicating Mc2 as functional analogs of 'pioneer myocytes' in developing mouse muscles, orchestrating the early phases of myogenesis and establishing a spatial and structural foundation for subsequent fusion events. These findings also highlight the intricate coordination between differentiation and fusion during primary myofiber development.

Finally, given the limited understanding of the regulation of *Mymx* expression, we investigated how its expression is downregulated in Mc2 cells. Previous studies have reported that MYOD1 binds the E-box motifs in the promoter region of *Mymx* (Zhang et al., 2020). We identified Mef2 and Rxrg as two of the most significant transcription factors during the Mc1 to Mc2 myocyte transition. The Mef2 family of genes, including *Mef2a*, has been implicated in muscle differentiation and regeneration (Liu et al., 2014). Similarly, Rxrg is known to play a role in the terminal differentiation of myoblasts (Zhu et al., 2009). We propose a novel regulatory model in which Mef2a is initially expressed and activates *Mymx* transcription, followed by Mef2a downregulation and Rxrg upregulation, which represses *Mymx* expression (Fig. 7).

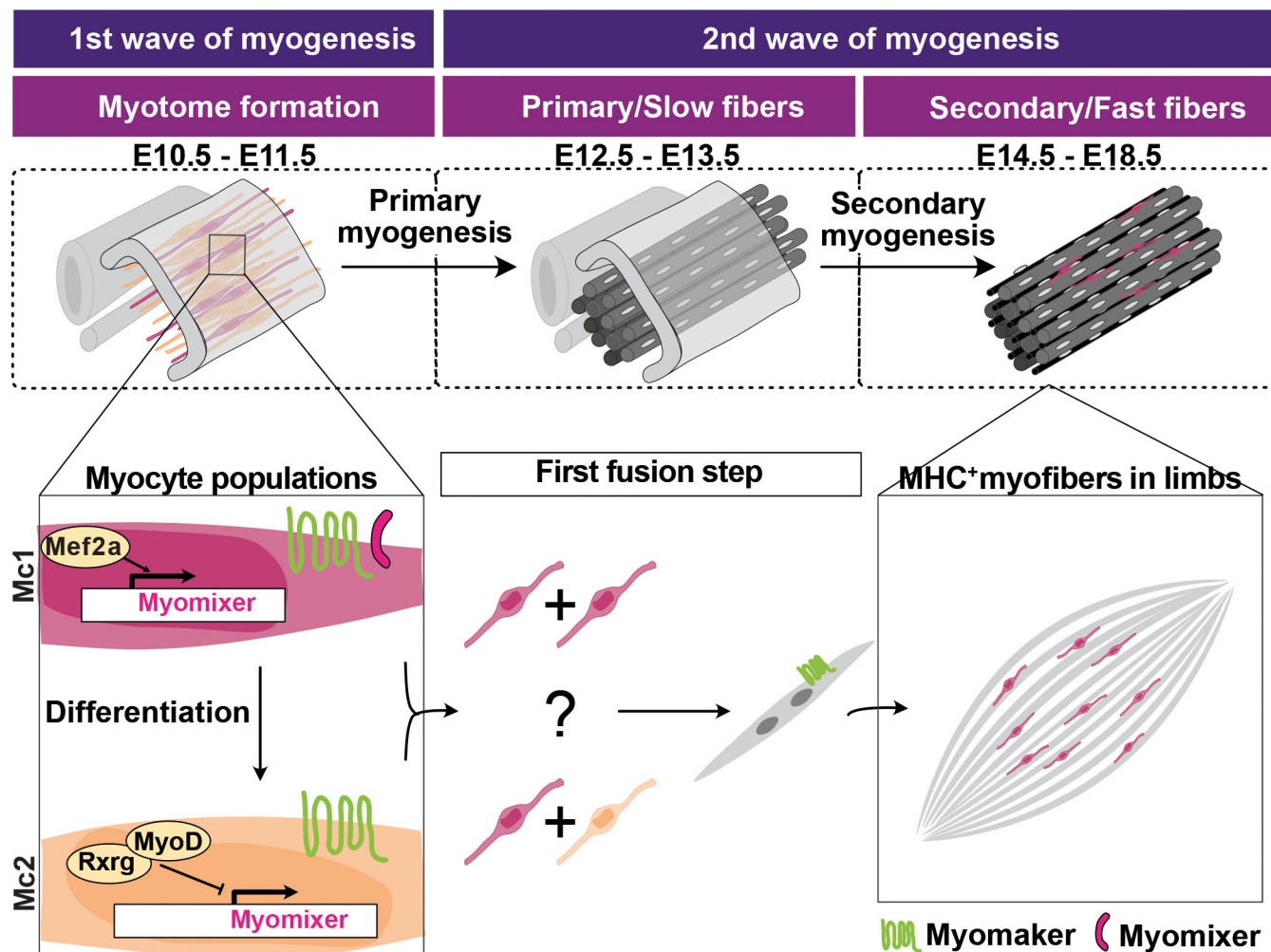

**Fig. 7. Schematic of the proposed mechanism of myogenesis in the mouse embryo.** During the first wave of myogenesis, and before the start of myocyte fusion, muscle progenitors form a heterogeneous population composed of the Myocyte 1 (Mc1) and the Myocyte 2 (Mc2). The Mc1, expressing myomixer, differentiate into the Mc2 downregulating myomixer. In the Mc1, Mef2a is expressed and is implicated in the positive regulation of *Mymx* transcription. In the Mc2, Rxrg is expressed and is implicated in the negative regulation of *Mymx*. During the second wave of myogenesis, muscle fibers are formed and only Mc1 are maintained. They reside in between the myofibers and preferably in the central part of the limb muscles. It remains to be determined whether the first fusion step is exclusively asymmetric (Mc1 to Mc2) or whether Mc1 to Mc1 fusion takes place *in vivo*.

Supporting evidence from a zebrafish study revealed that *Mymx* is expressed in fast myoblasts, but not in slow myoblasts, while *Mymk* was expressed in both populations (Yong et al., 2024). Although zebrafish and mammalian myogenesis share similarities, they diverge in key aspects, such as the specific organization of muscle precursors within the zebrafish somites generating slow fibers that are radially arranged around fast fibers. In contrast, the mammalian fiber types are established at later stages of development and are organized in a mixed pattern. Thus, findings from zebrafish cannot be fully extrapolated to the mammalian myoblast fusion. Based on our findings, we speculate that Mc2 play a role in regulating the number of muscle fibers per muscle. We propose that only a limited number of Mc1 cells differentiate into Mc2, thereby setting the foundation to organize primary muscle fibers.

Some questions remain to be explored. For example, how are the characteristics of a specific muscle conferred, and how is the number of fibers per muscle determined? Nevertheless, our findings provide a significant conceptual advancement in understanding embryonic myogenesis in vertebrates and should be considered when developing protocols to generate myogenic cells for treating muscle degeneration diseases.

## MATERIALS AND METHODS
### Single-cell RNA-sequencing
Pregnant C57BL/6 wild-type mice (*Mus musculus*) were euthanized at 9.5 days and 11.5 days post coitum (p.c.). Embryos were extracted and interlimb somites were dissected in a PBS, 1% bovine serum albumin (BSA) filled plate using Dumont #5SF forceps. Somites from three embryos of each embryonic age were aspirated with a wide-bore p200 pipette tip with the surrounding PBS and pooled together in low-bind 1.5 ml tubes. The pooled somites were centrifuged at 100 $g$ for 2 min at 4°C. Supernatant was aspirated and replaced with 50 µl trypsin (325-043-CL, Wisent Bioproducts) and incubated at 37°C for 15 min. Then 200 µl of fetal bovine serum (FBS; 090150, Wisent Bioproducts) was added, and cells were dissociated by pipetting up and down. Cells were centrifuged at 200 $g$ for 4 min at 4°C. Cells were washed twice with Dulbecco's Modified Eagle Medium (DMEM) supplemented with 10% FBS then manually counted (viability >90%) and filtered with a Flowmi 40 µm tip strainer (H13680-0040, Belart) and subsequently re-counted. Cells were adjusted at 1100 cells/µl and a 150 µl aliquot was sent for analysis using the 10x Chromium Chip (10x Genomics) for library preparation. Sequencing was performed with a High Output (150 cycles) flowcell using Illumina Nextseq 500. Sequence alignment was carried out using Cellranger count (v 3.0.1) with the reference genome mm10-1.2.0. For each sample (E9.5 and E11.5), a total of 10,000 cells were sequenced with ~30,000 median reads per cell and 3500 median genes per cell.

### Bioinformatic analyses
The Seurat (v 4.0.3) package under R (v4.0.5) (https://www.R-project.org/) was used to perform the clustering analysis on the scRNA-seq dataset (Stuart et al., 2019). The E9.5 and E11.5 datasets were integrated for comparison purposes and the clusters were manually annotated using publicly available databases such as the Mouse Organogenesis Cell Atlas (MOCA; https://oncoscape.v3.sttrcancer.edu.mouse.rna/landing). Only Fig. 1 was based on the integrated E9.5 and E11.5 datasets. For all subsequent figures, the E9.5 and E11.5 were analyzed independently, without using the integrated data. Within each dataset, data were initialized to keep only genes expressed in at least three cells and keep only cells that expressed at least 200 genes. Next, cells were filtered to eliminate duplicates and dying cells (nFeature_RNA>100 and <6500, percent.mt<15), the data were normalized (scale.factor=10,000), the VST method was used to find variable features, and the data were scaled and principal component analysis (PCA) analysis (npc=50) was performed. Next, the cell cycle phase scores were regressed out, and PCA was performed again. The Louvain algorithm was used to cluster the cells, using

18 dimensions and a resolution of 0.4. The clusters were visualized using the non-linear dimensional reduction method, UMAP. Clusters expressing muscle markers (*Pax3*, *Pax7*, *Myf5*, *Myod1*, *Myog* and *Myh3*) were identified, subsetted, normalized, re-scaled then re-clustered, taking into account 15 dimensions and a resolution of 0.4. The shown subclusters represent the direct output of the 'FindClusters()' function; no subclusters were merged or manually modified. The scCustomize (v2.1.2) package was used to generate custom violin and feature plots (Marsh et al., 2024). The Monocle3 (v1.0.0) package was used for differentiation trajectory prediction (Qiu et al., 2017). To perform the regulon analysis, we first generated a loom file using Velocyto (v0.17) from the 10x CellRanger output (La Manno et al., 2018). An anndata object was generated from the loom file and the muscle cells were subsetted based on the marker gene expression using scvelo (v0.2.3) (Bergen et al., 2020). The final anndata object was used as input to perform the regulon analysis using the pySCENIC (v 0.11.0) package (Aibar et al., 2017). CellRouter was used to predict key regulators contributing to cell state transitions occurring from Myoblast 1 to Myoblast 2. CellRouter is a trajectory inference tool implemented in the 'fusca' (v 1.0.0) package under R (v4.0.3). CellRouter calculates GRN scores to rank transcriptional factors based on their correlation with the pseudotime variable and their predicted targets (Lummertz da Rocha et al., 2018). The CellChat (version 1.6.1) R package was used for ligand-receptor cell communication prediction analysis (Jin et al., 2021).

### Fluorescence *in situ* hybridization
Mouse embryos at E9.5, E11.5 or limbs from E14.5 embryos were dissected and fixed for 15 min in 4% paraformaldehyde. They were then washed thrice with PBS, incubated for 2-4 h in 20% sucrose in PBS, and next embedded in OCT diluted 1:2 in 20% sucrose in PBS. The embryos were then sectioned at a thickness of 10 µm using a CryoStar NX70 (Thermo Fisher Scientific). Probe hybridization was performed using the reagents from the Multiplex fluorescent kit according to the protocol from the manufacturer (ACD Biotechne Brand). At the end of the protocol, the sections were further incubated with MF20 antibody [AB 2147781, Developmental Studies Hybridoma Bank (DSHB)] in PBS (1:10) for 1 h at 37°C and then with anti-mouse Alexa Fluor 488 (A-21200, Invitrogen) in PBS (1:500) for 30 min at room temperature. For the E9.5 embryos, we used probes against *Ifitm1* and *Myf5* RNAs. For the E11.4 and E14.5 embryos the probes used were against *Mymk* and *Mymx* RNAs. The stained E9.5 sections were imaged using the Leica Stellaris STED microscope. The stained E11.5 and E14.5 embryo sections were imaged using the Zeiss LSM710 confocal microscope.

### Immunofluorescence
Mouse embryos at E9.5, E11.5 or E14.5 embryos were dissected and fixed for 30 min or 1 h in 4% paraformaldehyde. They were then washed thrice with PBS, incubated for 2-4 h in 20% sucrose in PBS, and next embedded in OCT diluted 1:2 in 20% sucrose in PBS. Mouse embryo OCT sections (10 µm) were washed with PBS and blocked with 5% BSA in PBS supplemented with 0.01% saponin for 1 h at room temperature. Antibodies against Mymx (AF4580-SP, R&D Systems; 1:200), MF20 (AB 2147781, DSHB; 1:10) and Ki67 (Abcam, ab16667; 1:200) in the same blocking buffer were added and incubated overnight at 4°C. The slides were counterstained with donkey anti-mouse Alexa Fluor 488 (A-21200, Invitrogen; 1:1000), anti-sheep Alexa Fluor 568 (A-21099, Invitrogen; 1:500), and anti-rabbit Alexa Fluor 633 (A-21070, Invitrogen; 1:500). Images were acquired using a Zeiss LSM710 confocal microscope.

### Molecular cloning
For the luciferase reporter vectors, the highlighted promoters in Fig. S5 were cloned from a mouse genomic DNA preparation using primers listed in Table S3. The DNA fragments generated were then inserted using KpnI and XhoI restriction enzymes into the pGL4.10-Luc2 luciferase reporter plasmid (Promega). The clones were sequenced to verify the lack of mutations. For protein expression we used the MyoD-myc vector (Addgene plasmid #8399). MyoG-myc sequence (NM_0311892) flanked by attB sites was inserted into pDONR™221 vector via recombination. The gateway cloning technique was used to insert MyoG from the pDONR-MyoG vector into the pCS-pDEST-6Myc vector (Addgene plasmid #13070). For Rxrg-flag,

sequence (NM_009107.3) flanked by attB sites was inserted into pDONR™221 vector via recombination. The gateway cloning technique was used to insert Rxrg from the pDONR-Rxrg vector into the pcDNA-pDEST-FRT-3xflag-Cterm vector (Addgene plasmid #52505).

## Immunoprecipitation
HEK293T cells (tested negative for mycoplasma contamination) were transfected using polyethylenimine (23966-1, Polysciences) with the vectors expressing MyoD-myc, MyoG-myc and Rxrg-flag. Then, 48 h after transfection, cells were incubated with DSP solution at 1 mM [dithiobis(succinimidylpropionate), 22585, Thermo Fisher Scientific] for 30 min at room temperature to crosslink protein-protein interactions then with the stop solution (10 mM Tris, pH 7.5) for 15 min on ice to quench the reaction. Cells were lysed in NP40 lysis buffer (150 mM NaCl, 50 mM Tris, pH 7.5, 1% NP40) and 1 mg of proteins was incubated with 2 µg of mouse monoclonal myc-tag antibody (sc-40, Santa Cruz Biotechnology) bound on 40 µl of protein A agarose beads (15918-014, Invitrogen) at 4°C for 16 h. Beads were washed three times with NP40 lysis buffer, resuspended with Laemmli buffer (5% SDS, 0.1 mM Tris, pH 6.8, 140 mM β-mercaptoethanol, 25% glycerol) and heated at 95°C for 5 min to elute and denature the bound proteins. A quarter and three quarters of the elution were loaded on separate acrylamide gels for western blotting to look for the immunoprecipitated and the co-immunoprecipitated proteins, respectively. Membranes were incubated with a Rabbit myc-tag antibody (2272, Cell Signaling Technology; 1:1000) or anti-flag-HRP (A8592, Sigma-Aldrich; 1:10,000).

## Cell culture and RNA interference
C2C12 and HEK293T (tested negative for mycoplasma contamination) cells were maintained in DMEM supplemented with 20% and 10% fetal bovine serum (FBS), respectively, and 1% penicillin/streptomycin. C2C12 cells were plated in 12-well plates. Once they reached 90% confluency, they were switched to differentiation medium (DMEM supplemented with 2% horse serum and 1% penicillin/streptomycin) and transfected with 15, 25 or 50 ng of ON-TargetPlus siRNA SMARTpool for Mouse Rxrg (L-055799-01-0005, Dharmacon), Mef2a (L-041004-00-0005, Dharmacon) or Mef2c (L-041114-00-0005, Dharmacon) using the DharmaFECT1 reagent (T-2001-03, Dharmacon) following the manufacturer's protocol. For *Ifitm3* RNA interference, Qiagene GeneGlobe GS66141 (#1027416) was used. Cells were transfected again 48 h after the first transfection. After 96 h in the differentiation media, cells were lysed for RNA and protein analysis. For *ex vivo* cultures, somites we dissected from E11.5 embryos, dissociated and incubated overnight in DMEM supplemented with 20% FBS and 1% penicillin/streptomycin to adhere. They were next fixed and processed for immunostaining.

## RNA extraction and quantitative real-time PCR analysis
RNA was extracted using the Qiagen RNeasy minikit (74106). DNA was digested using DNase (79254, Qiagen). A total of 1 µg of RNA was reverse transcribed using the High-Capacity cDNA reverse transcription kit (4368814, Applied Biosystems). Quantitative real-time PCR was performed using PowerUp SYBRGreen Master mix (A25741, Applied Biosystems) in a ViiA7-96 machine. The delta-CT were normalized using the beta-actin levels. The primers used are listed in Table S3.

## Luciferase assay
C2C12 cells (tested negative for mycoplasma contamination) were plated in 24-well plates at a 50% confluence, transfected the same day using Lipofectamine 2000 reagent (11668019, Thermo Fisher Scientific) and incubated overnight. Triplicate wells were transfected with a total of 2 µg of DNA: Luciferase reporter plasmid (1 µg), the transcription factor of interest (between 15 and 500 ng) and the total was adjusted to 2 µg with an empty vector. The next day, the transfection media was replaced with media containing 1 µM 9-cis-retinoic acid (R4643, Sigma-Aldrich). Then, 36 h after transfection, cells were washed, lysis buffer was added (200 mM Tris, pH 7.5, 100 mM NaCl, 0.125% Triton) and cells were subjected to one freeze/thaw cycle. The lysate was used for the luminescence assay in a 96-well plate injected with the firefly luciferase assay buffer [100 mM Tris, pH 7.5, 10 mM DTT (DB0058, Bio Basic), 0.2 mM Coenzyme A (CAS 55672-92-9, Santa Cruz Biotechnology), 0.30 mM ATP (A7699-1G,

Sigma-Aldrich) and 2.8 mg/ml luciferin (E1605, Promega)]. The luminescence was measured using the Promega GloMax 96 Luminometer.

## Statistical analysis
Statistical tests were performed using GraphPad Prism 9.5.1 software. Details of the statistical tests used are provided in the corresponding figure legends. No data were excluded. For imaging and quantification, samples were anonymized.

## Generation of Mymx-T2A-Cre mouse line
For construction of the ssAAV2/1-Mymx-T2A-Cre donor plasmid, a commercially synthetized DNA fragment (GenScript) containing T2A-SV40NLS-Cre cassette flanked by homology arms (5′=200 bp and 3′=236 bp) has been inserted into ssAAV destination vector containing AAV2 ITRs using Gibson cloning. Production and purification of recombinant ssAAV2/1-Mymx-T2A-Cre vector have been performed by the Canadian Neurophotonics Platform Viral Vector Core Facility (RRID:SCR_016477). The Mymx-T2A-Cre knock-in mouse line was generated via CRISPR-Cas9 technology following the CRISPR-READI procedure, to have a T2A sequence and a Cre recombinase gene inserted immediately upstream of the translational stop codon of the *Mymx* gene (Chen et al., 2019).

Briefly, C57BL/6J zygotes were infected for 5 h in a 20 µl droplet of Advanced KSOM containing 3.6E12 GC/ml of ssAAV2/1-Mymx-T2A-Cre, at 37°C with 95% humidity and 5% $CO_2$. AAV-transduced zygotes were electroporated with pre-assembled RNPs containing sgRNA (Integrated DNA Technologies) at 9.6 µM and Alt-R S.p. HiFi Cas9 Nuclease V3 (Integrated DNA Technologies) at 8 µM (Chen et al., 2019). CRISPR-Cas9 guide RNA design tool from Integrated DNA Technologies were used to select gRNA. Treated zygotes were then surgically transferred into the oviducts of pseudopregnant CD1 females. Founder mice were identified by transgene-specific PCR and bred with wild-type C57BL/6J mice to establish the colony. Transgene-specific and junction PCRs followed by Sanger sequencing were performed on F1 mice. Experiments described were approved by the Animal Care Committee of the Institut de Recherches Cliniques de Montréal (protocol 2021-09 YK). The sequences of all primers used are listed in Table S3.

## Acknowledgements
We thank Dr M. Laurin (Université Laval) for critically reading the manuscript. We acknowledge the technical support of the IRCM Disease Modeling and Genome Editing facility (A. Gato) in generating the mutant mouse; the Microscopy facility (Drs D. Filion and M. Duguay); the Animal facility (S. Riverin). We thank the Molecular Biology and Functional Genomics facility (Drs O. Neyret and S. Boisel) for handling the single-cell RNA sequencing samples. We thank the Histology facility (S. Terouz) and our research assistant (A. Bourbia) for RNA scope (RNA *in situ* hybridization). We also thank the Bioinformatics platform (Dr Virginie Calderon) for preprocessing the RNA-seq data and generating the raw count matrix.

## Competing interests
The authors declare no competing or financial interests.

## Author contributions
Conceptualization: J.-F.C., S.N.; Data curation: S.N., A.J., L.J.; Formal analysis: S.N., A.J., J.S., K.K., F.L.G.; Funding acquisition: J.-F.C.; Investigation: S.N., F.L.G.; Methodology: S.N., Y.K., K.K.; Project administration: J.-F.C.; Resources: Y.K., Y.P., W.K., M.C.; Supervision: J.-F.C., M.M., M.C.; Visualization: S.N.; Writing – original draft: J.-F.C., S.N.; Writing – review & editing: J.-F.C., S.N., F.L.G., M.C.

## Funding
This work was supported by a grant to J.-F.C. from the Canadian Institutes of Health Research (PJT-153065). J.-F.C. holds the Canada Research Chairs (Tier 1) in Cellular Signaling and Cancer Metastasis, and the Alain Fontaine Chair in Cancer (from the Fondation de l'Institut de recherches cliniques de Montréal). S.N. was supported by a doctoral studentship from the Fonds de Recherche du Québec – Santé. Open Access funding provided by McGill University. Deposited in PMC for immediate release.

## Data and resource availability
The raw scRNA-seq data have been deposited at GEO under accession number GSE290276 (see also https://jfcote.shinyapps.io/shinyappmulti/). All relevant data and details of resources can be found within the article and its supplementary information.

**Peer review history**
The peer review history is available online at https://journals.biologists.com/dev/lookup/doi/10.1242/dev.204771.reviewer-comments.pdf

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
