## [Peer Review File · Development (Cambridge, England)]

Differentially expressed fusogens specify myocyte states to drive myogenesis

Sarah Nahlé, Awais Javed, Loïck Joumier, Yacine Kherdjemil, Julie Sitolle, Konstantin Khetchoumian, Yash Parekh, Wojciech Krezel, Mohan Malleshaiah, Fabien Le Grand, Michel Cayouette and Jean-François Côté

DOI: 10.1242/dev.204771

Editor: Benoit Bruneau

Review timeline

Original submission:	10 March 2025
Editorial decision:	22 April 2025
First revision received:	5 August 2025
Accepted:	2 September 2025

Original submission

First decision letter

MS ID#: dev.204771

MS TITLE: Differentially expressed fusogens specify myocyte states to drive myogenesis

AUTHORS: Jean-François Côté; Sarah Nahlé; Awais Javed; Loïck Joumier; Yacine Kherdjemil; Konstantin Khetchoumian; Yash Parekh; Wojciech Krezel; Mohan Malleshaiah; Fabien Le Grand; Michel Cayouette

Dear Dr Côté,

I have now received all the referees' reports on the above manuscript, and have reached a decision. The referees' comments are appended below, or you can access them online: please go to:

As you will see, the referees express considerable interest in your work, but have some significant criticisms and recommend a substantial revision of your manuscript before we can consider publication. If you are able to revise the manuscript along the lines suggested, which may involve further experiments, I will be happy receive a revised version of the manuscript. Your revised paper will be re-reviewed by one or more of the original referees, and acceptance of your manuscript will depend on your addressing satisfactorily the reviewers' major concerns. Please also note that Development will normally permit only one round of major revision. If it would be helpful, you are welcome to contact us to discuss your revision in greater detail. Please send us a point-by-point response indicating your plans for addressing the referees' comments, and we will look over this and provide further guidance.

Please attend to all of the reviewers' comments and ensure that you clearly highlight all changes made in the revised manuscript. Please avoid using 'Tracked changes' in Word files as these are lost in PDF conversion. I should be grateful if you would also provide a point-by-point response detailing how you have dealt with the points raised by the reviewers in the 'Response to Reviewers' box. If you do not agree with any of their criticisms or suggestions please explain clearly why this is so.

Reviewer 1

In this manuscript, Nahle et al. have used scRNA-seq to examine the formation of multinucleated myofibers in the developing mouse embryo. Using somites from the E9.5 and E11.5 embryos, they identified four myogenic cell populations that include cycling myoblasts, myoblasts, and muscle precursors. The spatial resolution of transcripts showed a population of early myogenic cells expressing the transcript *lftim*. Using C2C12 cells, they show that *lftim3* acts as an inhibitor of cell fusion. Further clustering of the myogenic populations identified two myocyte populations -one that expresses *Mymk* and *Mymx* (*mc1*) while the second expresses only *Mymk* (*mc2*). Lineage tracing showed the *Mc2* population is derived from the *Mc1* population. Using bioinformatics to identify regulons expressed in *Mc1* and *Mc2*, *Mef2a* was identified as an activator of *Mymx* expression while *Rxrg* was identified as a repressor of *Mymx*. Based on these findings, the authors conclude that *Mc1* and *Mc2* represent myogenic cell populations that drive primary and secondary myofiber formation, a two cell-type system of myofiber formation analogous to the founder cells and fusion competent myoblasts in *Drosophila*.

This manuscript reports the novel finding that two different myocyte populations exist in somatic regions of the limb during muscle development. The identification of fusion competent cells that express myomaker but not myomerger was not expected based on the current understanding of cell fusion, and represents an important advance in the field. The existence of these two distinct cell populations identified by scRNA-Seq was confirmed at the protein level using immunofluorescence in sections from the embryo, solidifying this important finding. The experiments looking at the mechanisms driving the transition between the *Mymx* positive and negative populations represent a good starting point for examining the regulated transition between cell populations, however they fail to provide strong evidence that the transition is driven by *Rxrg* blocking *MyoD/Mef2a* activity. Overall this is an exciting advance for the field, but additional studies would be needed to solidify the transcriptional regulation of the transition between cell states.

Issues to be addressed:

1. The authors have concluded that *RXRg* represents a key regulator of the transition between *Mc1* and *Mc2* based on the repressed expression of *Mymx* but not *Mymk*. This is consistent with Scenic analysis suggesting that *RXRg* is a key transcription factor in *Mc2* cells. However, the model in Figure 7 proposes that *Mef2a* would drive *Mymx* expression in *Mc2* and competition between *MyoD/RXRg* would lead to a loss of *Mymx*, a model where *RXRg* acts as a repressor to lead to a population transition. This reviewer would suggest that the transcriptional mechanisms might be too complicated to resolve with a short promoter fragment transfected into C2C12 cells. scRNA-seq suggests that *MyoD* and *Mef2a* are down-regulated in the transition from *Mc1* to *Mc2*. The *RXRg* regulon then becomes more important in *Mc2*. Given that *MyoD* and *Mef2a* likely work together to activate *Mymx* in *Mc1* cells, it is unclear why *Mef2a* was not included in the transfections with *MyoD/Myog* and *RXRg*. The luciferase assays should be complemented with experiments examining endogenous mRNA by RT-qPCR after knock-down and/or overexpression of combinations of *MyoD/Mef2a/RXRg* as was done in figure 6F.
2. *Rxrg* is thought of as a repressor when unliganded, and an activator when in the presence of 9-cisRA. Is the blocking of *MyoD/Myogenin* activity by *RXRg* dependent on its binding to ligand?
3. The lineage tracing experiments are weakened by the lack of TdT expression in *Mc1* cells. In lines 316-318 of the text, it is proposed that the lack of TdT signal in the *Mc1* cells may be due to the rapid fusion of these cells after generation. This does not seem probable since time points of E11.5 and E14.5 have only *Mc1* cells, and these cells are deemed to be proliferative based on Ki-67 staining. Wouldn't the continued proliferation of *Mc1* cells lead to continued expression of the TdT. Perhaps the *Rosa26* locus is less expressed in *Mc1* compared to *Mc2*. Can the TdT transcript be detected in the *Mc1* cells?
4. The figure legend for figure 5 is missing.

Reviewer 2

SUMMARY OF THE ADVANCE MADE IN THIS PAPER AND ITS POTENTIAL SIGNIFICANCE TO THE FIELD

In this manuscript the authors perform scRNAseq of somites during embryonic development and find multiple myogenic clusters. They unexpectedly found myocyte populations that differed in the expression of *Mymk* and *Mymx*, which are factors required for myocyte fusion. Though expression analysis (protein and mRNA) and lineage tracing, they propose that *Mymk*⁺ *Mymx*⁺ cells transition to *Mymk*⁺ *Mymx*⁻ cells (that are more differentiated). They also identify that *Mef2* and *Rxr* transcriptionally control *Mymx* expression. These results are very interesting for the field and the manuscript is well-written. However, I am concerned that there are major over-interpretations of scRNAseq data and expression data in a small number of samples that are not quantified. Specifically, the interpretation that *Mymk*⁺ *Mymx*⁺ cells fuse with *Mymk*⁺ *Mymx*⁻ cells is not justified and there should be a discussion of alternative possibilities and models. The claims related to transcriptional control of *Mymx* expression by *Mef2* and *Rxr* are supported by the data.

SUGGESTIONS TO AUTHORS

1. Is the localization of *Ifitm* and *Myf5* from 2E quantified?
2. One over-interpretation is whether *Ifitm3* needs to be down-regulated for fusion. Knockdown in C2C12s increases both differentiation and fusion making it difficult to conclude which process is impacted. There is also no data of the magnitude of knockdown with the siRNAs. For *Ifitm* expression in myogenic populations from scMuscle Atlas (Figure S2A) it seems *Ifitm3* is expressed in myocytes, which should be fusogenic, so in this situation it is not down-regulated? Myonuclei would be already fused and *Ifitm3* is down-regulated there.
3. The details surrounding scRNAseq data processing are not clear. Line 148-153: How were these nuclei subclustered? Did the authors subset from the integrated object, and rerun dimensionality reduction techniques? A cleaner approach less susceptible to technical variation or batch effects between the datasets would be to subcluster directly from the E9.5 dataset, instead of running on the integrated assay. Were variable features found between these subset myogenic progenitors to run clustering on, or were these clustered based on the variable features of the entire integrated object? It's hard to interpret these clusters given no clear guidance on how the single cell data was processed. When four subclusters were identified, was that the raw output of the `FindClusters()` function with default parameters? Were certain sub clusters merged?
4. The reason to remove cluster 7 in Figure 2A is not clear. Cells with low total RNA and gene counts should be removed during original processing, which means those cells passed minimal quality control metrics.
5. Line 211: This seems like a broad overstatement. Based on the data, one could suggest the following title for the section- Expression of sarcomeric proteins defines distinct myocyte subpopulations at E11.5. This could be less biased and probably a bit more accurate (See authors own points at lines 226-227).
6. Line 238: Following this logic in the presented heatmap, *Mymx* expression seems to persist from MC1 to MC2, with myoblast 2 *Mymx*⁺ cells present in the cluster. In fact, there seems to be *Mymx*⁺ and *Mymk*⁻ cells present in the myoblast 2 population (Figure 2F).
7. Line 245-250: Authors state here *Mymk* expression is in both populations, while *Mymx* is only present in the earlier MC1 population. *Mymx* could be harder to detect, and more "fusion state" myocytes would likely cluster with earlier differentiated myocytes, while post fusion myocytes that may retain the more profiled *Mymk* transcripts would be clustering more on expression of *myh3* and sarcomeric transcripts.
8. Are MC2 cells fused? It looks like multiple Mc2 cells (*Mymk*⁻) in Figure 3G and 3H have at two nuclei. Since the authors state that these analyses can't be quantified, I don't know how they can exclude the simple possibility is that *Mymx* is down-regulated after fusion. The idea of down-regulation of *Mymx* after fusion is also consistent with the absence of Mc2 cells later in embryonic development (E14.5) and this may be what the authors are saying in Lines 292-294. For the dissociation in Figure S4, what were *Mymk* (protein/mRNA) in these cells? The lack of quantification is an issue to know whether these events are standard or rare?

9. For Figure 3C, This heat map does not suggest a Mymk+/Mymx- cluster. MC1 cluster is certainly enriched for MYMX, but it would be helpful to show the percentage of Mymx expression in the two clusters. Can the authors please provide violin plots for mymx and myomaker expression? The point sizes of the feature plots, seem to obscure positive Mymx cells in the MC2 cluster. Additionally, no relative expression color scaling for the feature plots, so are these all cells with at least one count of Mymx or Mymk? A violin plot will better highlight enrichment in both intensity of expression and frequency of expression of these two genes in these populations.

10. The authors assess Mc1 and Mc2 populations during secondary myogenesis, which they indicate takes place between E14.5 and E18.5. Are they defining secondary myogenesis strictly based on age of the embryo or the cell fusion events occurring in muscle where primary is fusion of myocytes to form a new myofiber and secondary is fusion of myocytes to myofibers?

11. Line 318: I am worried about the interpretation that Mc1 rapidly fuses with either an Mc2 or a myofiber in proximity. This was state in relation to the Mymx-Cre data. For the Mymx-Cre allele, the idea that Mc1 cells are not labelled at E11.5 due to a lag of Cre expression and recombination is possible. It's unclear why these data could be related to rapid fusion events. Aren't Mc1 cells around during embryogenesis, so they don't rapidly fuse? Are the Mc1 cells between E14.5 and E18.5 Tom+?

12. What are the data that a Mc1 cell can't fuse with another Mc1 cell? For the model in Figure 4G, I don't see any direct data that Mc1 cells fuse to Mc2 cells. I would strongly suggest the authors remove the model. If the down-regulation of Mymx is necessary for fusion as they predict why would this process be used at other times of fusion from E14-E18

First revision

Author response to reviewers' comments

We would like to sincerely thank the editor and the reviewers for their careful evaluation of our manuscript. We greatly appreciate the time and effort that went into the review process, as well as the constructive comments and suggestions provided. ***These insights have been invaluable in helping us strengthen the manuscript.*** Below, we provide detailed responses to each comment. Reviewer comments are shown in bold, followed by our responses.

REVIEWER 1

In this manuscript, Nahle et al. have used scRNA-seq to examine the formation of multinucleated myofibers in the developing mouse embryo. Using somites from the E9.5 and E11.5 embryos, they identified four myogenic cell populations that include cycling myoblasts, myoblasts, and muscle precursors. The spatial resolution of transcripts showed a population of early myogenic cells expressing the transcript Ifitm. Using C2C12 cells, they show that Ifitm3 acts as an inhibitor of cell fusion. Further clustering of the myogenic populations identified two myocyte populations -one that expresses Mymk and Mymx (mc1) while the second expresses only Mymk (mc2). Lineage tracing showed the Mc2 population is derived from the Mc1 population. Using bioinformatics to identify regulons expressed in Mc1 and Mc2, Mef2a was identified as an activator of Mymx expression while Rxrg was identified as a repressor of Mymx. Based on these findings, the authors conclude that Mc1 and Mc2 represent myogenic cell populations that drive primary and secondary myofiber formation, a two cell-type system of myofiber formation analogous to the founder cells and fusion competent myoblasts in drosophila.

This manuscript reports the novel finding that two different myocyte populations exists in somatic regions of the limb during muscle development. The identification of fusion competent cells that express myomaker but not myomerger was not expected based on the current understanding of cell fusion, and represents an important advance in the field.

The existence of these two distinct cell populations identified by scRNA-Seq was confirmed at the protein level using immunofluorescence in sections from the embryo, solidifying this important finding. The experiments looking at the mechanisms driving the transition between the Mymx positive and negative populations represent a good starting point for examining the regulated transition between cell populations, however they fail to provide strong evidence that the transition is driven by Rxrg blocking MyoD/Mef2a activity. Overall this is an exciting advance for the field, but additional studies would be needed to solidify the transcriptional regulation of the transition between cell states.

We thank the reviewer for their positive evaluation of our work of for highlighting the potential impact of our findings.

1. *Issues to be addressed:*

The authors have concluded that RXRg represents a key regulator of the transition between Mc1 and Mc2 based on the repressed expression of Mymx but not Mymk. This is consistent with Scenic analysis suggesting that RXRg is a key transcription factor in Mc2 cells. However, the model in Figure 7 proposes that Mef2a would drive Mymx expression in Mc2 and competition between MyoD/RXRg would lead to a loss of Mymx, a model where RXRg acts as a repressor to lead to a population transition. This reviewer would suggest that the transcriptional mechanisms might be too complicated to resolve with a short promoter fragment transfected into C2C12 cells. scRNA-seq suggests that MyoD and Mef2a are down-regulated in the transition from Mc1 to Mc2. The RXRg regulon then becomes more important in Mc2. Given that MyoD and Mef2a likely work together to activate Mymx in Mc1 cells, it is unclear why Mef2a was not included in the transfections with MyoD/Myog and RXRg. The luciferase assays should be complemented with experiments examining endogenous mRNA by RT-qPCR after knock-down and/or overexpression of combinations of MyoD/Mef2a/RXRg as was done in figure 6F.

We appreciate the comments of the reviewer, and we fully agree that a minimal promoter approach will not be sufficient to resolve the full regulation. In fact, we are quite interested by the regulons that we identified and further studies by us and others will be needed in the future. At this time, our goal was to define at least some of the regulation at play that may contribute to the Mc1 to Mc2 transition, without the pretention to clarify everything. As such, in Fig. 7, we attempted to summarize two findings:

1. Mef2a acts as a positive regulator of *Mymx* transcription in Mc1 cells.
2. RXRg interacts with MyoD, as confirmed by immunoprecipitation data, leading to the inhibition of *Mymx* transcription.

We do not propose a competitive mechanism. To eliminate any ambiguity, we have revised Figure 7 accordingly. Instead, our model delineates a distinct regulatory pathway: *Mef2a* is initially expressed in Mc1 cells, where it promotes *Mymx* expression. Subsequently, in Mc2 cells, *Mef2a* is specifically downregulated, and a complex formed by RXRg and MyoD contributes to the negative regulation of *Mymx* expression. This sequential regulation underscores a dynamic shift in transcriptional control rather than a competitive interaction. The relevant portion of the revised figure is shown below.

We thank the reviewer for suggesting an additional co-expression experiment. Our rationale for not including *Mef2a* in the transfection with *MyoD* and *RXRg* is based on their expression dynamics. Specifically, *Mef2a* and *RXRg* are not co-expressed, rather, they exhibit a mutually exclusive expression pattern in *Mc1* and *Mc2*, respectively. Careful examination of our scRNA-seq confirms that these two genes are almost never expressed in the same cells, as illustrated in the blended feature plot shown below. The reviewer's comment made us realize that we may not have been sufficiently clear about this point in the original version of the paper. Thus, we have now clarified the Discussion that our model does not propose a competitive interaction between *Mef2a* and *RXRg*, but rather a sequential cascade of induction and inhibition events.

2. *Rxrg* is thought of as a repressor when unliganded, and an activator when in the presence of 9-cisRA. Is the blocking of *MyoD*/*Myogenin* activity by *RXRg* dependent on its binding to ligand?

We appreciate the reviewer's insightful comment, which aligns with considerations we explored during our study. To address this, we performed luciferase reporter assays using the *Mymx* promoter in the presence or absence of 9-cis retinoic acid (9cRA). These experiments demonstrated that *Rxrg*-mediated transcriptional repression of *Mymx* is independent of 9cRA. Specifically, *Rxrg* consistently repressed *MyoD*-induced *Mymx* expression regardless of 9cRA treatment. While we observed a slight trend toward enhanced repression in the presence of 9cRA (figure below, for reviewers only), the effect was modest and did not justify further mechanistic investigations. We included this clarification in the Discussion.

3. The lineage tracing experiments are weakened by the lack of TdT expression in Mc1 cells. In lines 316-318 of the text, it is proposed that the lack of TdT signal in the Mc1 cells may be due to the rapid fusion of these cells after generation. This does not seem probable since time points of E11.5 and E14.5 have only Mc1 cells, and these cells are deemed to be proliferative based on Ki-67 staining. Wouldn't the continued proliferation of Mc1 cells lead to continued expression of the TdT. Perhaps the Rosa26 locus is less expressed in Mc1 compared to Mc2. Can the TdT transcript be detected the Mc1 cells?

We thank the reviewer for this important comment regarding a key experiment of our study. We acknowledge that our previous statement concerning cell fusion was likely inaccurate, as the reviewer suggested, and we have therefore removed it from the manuscript. While a subset of Mc1 cells remain proliferative, *Mymx* expression is newly initiated at E11.5. Thus, we maintain our interpretation that a delay between Cre expression and effective recombination may account for some of the observed timing of expression. This phenomenon has also been demonstrated by other groups, as referenced in the manuscript. This clarification has been incorporated into the revised manuscript.

To directly address the valid concern raised by Reviewer 1, and a similar point from Reviewer 2, we conducted additional experiments later in development, focusing on Mc1 cells during secondary myogenesis. In E14.5 *Mymx*^{T2A-Cre/+}-nls-tdTomato^{-/+} embryos, we observed Mc1 cells labeled with nls-tdTomato, whereas *Mymx*^{WT}-nls-tdTomato^{-/+} embryos lacked this fluorescent signal, as predicted. We found that nls-tdTomato labeling was present in myonuclei withing MHC+ myofibers, demonstrating that they originate from the Mc1 lineage. We believe that these results directly address the reviewers' primary concerns. The new data is added to Figure 5 and is shown below.

4. The figure legend for figure 5 is missing.

Thank you for pointing that out. We have corrected this.

REVIEWER 2

SUMMARY OF THE ADVANCE MADE IN THIS PAPER AND ITS POTENTIAL SIGNIFICANCE TO THE FIELD

In this manuscript the authors perform scRNAseq of somites during embryonic development and find multiple myogenic clusters. They unexpectedly found myocyte populations that differed in the expression of Mymk and Mymx, which are factors required for myocyte fusion. Though expression analysis (protein and mRNA) and lineage tracing, they propose that Mymk+ Mymx+ cells transition to Mymk+ Mymx- cells (that are more differentiated). They also identify that Mef2 and Rxr transcriptionally control Mymx expression. These results are very interesting for the field and the manuscript is well-written. However, I am concerned that there are major over-interpretations of scRNAseq data and expression data in a small number of samples that are not quantified. Specifically, the interpretation that Mymk+ Mymx+ cells fuse with Mymk+Mymx- cells is not justified and there should be a discussion of alternative possibilities and models. The claims related to transcriptional control of Mymx expression by Mef2 and Rxr are supported by the data.

We thank this reviewer for their positive assessment of our study, highlighting that the results are of great interest to the field. We also appreciate the deep reading of the manuscript and the numerous constructive comments to improve the quality of the manuscript and decrease over-interpretation of some data.

SUGGESTIONS TO AUTHORS

1. Is the localization of Ifitm and Myf5 from 2E quantified?

In response to this comment, we quantified the localisation of *Ifitm1* and *Myf5* expression as visualized by RNAscope (Figure 2E). To support this, we generated the figure below and included as part of Figure S2 in the revised manuscript. Quantification was performed by counting RNA puncta per somite using Imaris software. The numbering of somites was assigned empirically: somite number 1 is the first somite present in the anterior end of the embryo section. We prefer not to show the pooled graph (below) in the paper because somite 1 in embryo 1 is not directly comparable to somite 1 in the other two embryos. Nevertheless, the trend is consistent across all samples: *Myf5* expression decreases along the anterior-posterior axis, while *Ifitm1* expression increases posteriorly.

2. One over-interpretation is whether *Ifitm3* needs to be down-regulated for fusion. Knockdown in C2C12s increases both differentiation and fusion making it difficult to conclude which process is impacted. There is also no data of the magnitude of knockdown with the siRNAs. For *Ifitm* expression in myogenic populations from scMuscle Atlas (Figure S2A) it seems *Ifitm3* is expressed in myocytes, which should be fusogenic, so in this situation it is not down-regulated? Myonuclei would be already fused and *Ifitm3* is down-regulated there.

We thank the reviewer for this insightful comment. We agree that the observation that *Ifitm3* knockdown increases both differentiation and fusion introduces complexity in interpreting the data. As requested, we now provide quantitative data demonstrating efficient knockdown of *Ifitm3* using two different siRNAs, each achieving approximately around 60% reduction in expression. These results are now included Figure S2 (and below).

Importantly, while *Ifitm3* knockdown does result in a statistically significant increase in differentiation, the magnitude of this effect is relatively modest. In contrast, the impact on fusion is more pronounced: we observe an increase of over 10% in fusion index under knockdown conditions with both siRNAs (Figure 2F). Notably, nearly 50% of myotubes in the knockdown conditions have 11 or more nuclei, compared to 15% in the control conditions. Given these findings, we have revised the manuscript to state that *Ifitm3* knockdown likely promotes fusion, while adopting a cautious tone to reflect the potential confounding effect of increased differentiation.

In response to the comment on Fig. S2B, we have replaced the panel with a new and clearer representation of the dataset illustrating the dynamic expression of IFITM genes during muscle regeneration. The revised Fig. S2B demonstrates that *Ifitm2* and *Ifitm3* are expressed in quiescent and activated muscle stem cells but progressively decline as differentiation advances. Notably, their expression is markedly reduced in fusion competent myocytes compared to less differentiated progenitor cells, supporting their roles as “fusion inhibitors” in early stages of myogenic activation.

3. The details surrounding scRNAseq data processing are not clear. Line 148-153: How were these nuclei subclustered? Did the authors subset from the integrated object, and rerun dimensionality reduction techniques? A cleaner approach less susceptible to technical variation or batch effects between the datasets would be to subcluster directly from the E9.5 dataset, instead of running on the integrated assay. Were variable features found between these subset myogenic progenitors to run clustering on, or were these clustered based on the variable features of the entire integrated object? It's hard to interpret these clusters given no clear guidance on how the single cell data was processed. When four subclusters were identified, was that the raw output of the FindClusters() function with default parameters? Were certain sub clusters merged?

We thank the reviewer for their constructive comment. We agree that providing additional details on our data analysis approach is important to improve clarity of the manuscript. To address this, we have added a more detailed description in the Methods section (highlighted in yellow below). Briefly, to clarify: 9.5/11.5 integrated data was used exclusively for the analyses presented in Figure 1. All subsequent analyses throughout the manuscript were performed on either the E9.5 or the E11.5 datasets, which were analyzed separately, as suggested by the

reviewer.

“The Seurat (v 4.0.3) package under R (v4.0.5) (R Core Team (2024) was used to perform the clustering analysis on the scRNA-seq dataset ⁶². The E9.5 and E11.5 datasets were integrated for comparison purposes and the clusters were manually annotated using publicly available databases such as Discovery LifeMap (<https://discovery.lifemapsc.com/>) and the Mouse Organogenesis Cell Atlas (MOCA) (<https://oncoscape.v3.sttrcancer.org/atlas.gs.washington.edu.mouse.rna/landing>). Only Figure 1 was based on the integrated E9.5 and E11.5 datasets. For all subsequent figures, the E9.5 and E11.5 were analysed independently, without using the integrated data. Within each dataset, data was initialized to keep only genes expressed in at least 3 cells and keep only cells that express at least 200 genes. Next, cells were filtered to eliminate duplicates and dying cells ($n_{\text{Feature_RNA}} > 100$ and < 6500 , $\text{percent.mt} < 15$), the data was normalized ($\text{scale.factor} = 10000$), the VST method was used to find variable features, and the data was scaled and PCA analysis ($\text{npc} = 50$) was performed. Next, the cell cycle phase scores were regressed out, and PCA was performed again. The Louvain algorithm was used to cluster the cells, using 18 dimensions and a resolution of 0.4. The clusters were visualized using the non-linear dimensional reduction method, UMAP. Clusters expressing muscle markers (*Pax3*, *Pax7*, *Myf5*, *Myod1*, *Myog* and *Myh3*) were identified, subsetted, normalized, re-scaled then re-clustered, taking into account 15 dimensions and a resolution of 0.4. The shown subclusters represent the direct output of *FindClusters()* function; no subclusters were merged or manually modified. (...)”

4. The reason to remove cluster 7 in Figure 2A is not clear. Cells with low total RNA and gene counts should be removed during original processing, which means those cells passed minimal quality control metrics.

This is a pertinent comment from the reviewer. As detailed in the manuscript, Cluster 7 in Figure 3A is likely a technical artifact (note: we assumed the reviewer refers to figure 3A. If not, Fig 2A, which is the analysis of the separate E9.5 dataset, does not contain a cluster 7). We thoroughly validated our bioinformatic pipeline, and the standard threshold filtering and quality controls (QC) were applied to exclude cells with high RNA content (typically indicative of doublets or multiplets) or elevated mitochondrial RNA levels (often associated with stressed or dying cells). Conversely, cells with low total RNA and mitochondrial RNA counts are generally retained, as they may represent normal, quiescent or small cells. However, when such cells fail to express specific marker genes, as observed in Cluster 7, they may in fact be empty droplets contaminated with ambient RNAs. In our case, we strongly suspect the ambient RNAs originate from the complex initial material: dissociated embryonic tissues containing developing muscle. It is plausible that a small number of bi-nucleated cells lysed during processing, possibly in the microfluidic chip, contributing to this contamination.

That said, we tried filtering out cells with lower counts (see below). Cluster 7 was eliminated, but importantly, the remaining clustering and biological interpretations were unchanged compared to the original analysis.

However, since the original analysis was conducted 4 years ago, the top differentially expressed genes (DEGs) in the heatmap (Figure 3B) have shifted slightly due to updates in software versions and methods. Key genes such as *Myod1*, *Myog* and *Pax3* are still present among the DEGs, but they no longer rank in the top 10 genes, making the heatmap less representative of the biology.

Therefore, we propose to retain the original UMAP as it better captures the key biological markers relevant to the study. The rest of the figures remain unaffected by the updated filtering. We clarified in the manuscript that cluster 7 is a technical artifact of no biological significance.

5.Line 211: *This seems like a broad overstatement. Based on the data, one could suggest the following title for the section- Expression of sarcomeric proteins defines distinct myocyte subpopulations at E11.5. This could be less biased and probably a bit more accurate (See authors own points at lines 226-227).*

We thank the reviewer for this constructive comment. We agree that the original title, which emphasized *Mymx* alone, may have overstated its role given the broader set of differentially expressed genes (DEGs) distinguishing Mc1 and Mc2. To better reflect the complexity of our findings while still highlighting the novel and unexpected observation that *Mymx* is differentially expressed among myocytes, a key contribution of our study, we have revised the section title. Our new title, “Myomixer expression is lacking in a subpopulation of myocytes at E11.5”, strikes a balance between accuracy and emphasis on the novel aspect.

6.Line 238: *Following this logic in the presented heatmap, Mymx expression seems to persist from MC1 to MC2, with myoblast 2 Mymx+ cells present in the cluster. In fact, there seems to be Mymx+ and Mymk- cells present in the myoblast 2 population (Figure 2F).*

This is a thoughtful observation from this reviewer. It is indeed correct that *Mymx* RNA remains detectable in Mc2 cells, as shown in the heatmap in Figure 3D. However, we interpret this signal in the context of a progressive Mc1-to-Mc2 transition, which reflects a continuum of myogenic differentiation. In such a dynamic process, it is expected that transcripts like *Mymx*, which are actively downregulated, may still be transiently present in cells that are transitioning between states. Importantly, our immunofluorescence data show that *Mymx* protein is not detected in Mc2 cells (MHC⁺), supporting the conclusion that the residual *Mymx* RNA signal observed in the scRNA-seq data likely reflects ongoing transcriptional downregulation rather than sustained functional expression. This distinction between RNA presence and protein expression is critical, and we believe it strengthens our interpretation that *Mymx* is effectively absent at the protein level in Mc2 cells, despite low-level RNA presence.

7.Line 245-250: Authors state here Mymk expression is in both populations, while Mymx is only present in the earlier MC1 population. Mymx could be harder to detect, and more "fusion state" myocytes would likely cluster with earlier differentiated myocytes, while post fusion myocytes that may retain the more profiled Mymk transcripts would be clustering more on expression of myh3 and sarcomeric transcripts.

We thank the reviewer for this thoughtful interpretation. However, our data strongly support the conclusion that Mc2 are mononucleated myocytes, not post-fusion cells:

- RNA content and QC filtering: fused myocytes would be expected to contain approximately double the RNA content of mono-nucleated cells. Such cells were excluded during our quality control filtering, which removed outliers with abnormally high RNA content.
- Microfluidic constraints: the microfluidic chip used for single-cell capture (50-60 um channels) is unlikely to accommodate bi-nucleated or larger fused myocytes. These cells would likely lyse during processing, potentially generating debris-like profiles such as those observed in Cluster 7.
- Immunofluorescence validation: both in embryo sections and in dissociated somite preparations, Mc2 cells were predominantly mono-nucleated and did not express *Mymx*, consistent with our scRNA-seq findings. This is further quantified and discussed in response to comment 8, below.

8.Are MC2 cells fused? It looks like multiple Mc2 cells (Mymk-) in Figure 3G and 3H have at two nuclei. Since the authors state that these analyses can't be quantified, I don't know how they can exclude the simple possibility is that Mymx is down-regulated after fusion. The idea of down-regulation of Mymx after fusion is also consistent with the absence of Mc2 cells later in embryonic development (E14.5) and this may be what the authors are saying in Lines 292-294. For the dissociation in Figure S4, what were Mymk (protein/mRNA) in these cells? The lack of quantification is an issue to know whether these events are standard or rare?

We appreciate the reviewer's thoughtful comment and agree that *Mymx* downregulation after fusion is potentially valid. However, our data support that the majority of terminally differentiated Mc2 cells are mononucleated and lack *Mymx* expression, which is consistent with their identity as unfused myocytes.

To address the marker clarification issue, we explain in the discussion that Mc2 were identified using MHC (Myosin Heavy Chain) as a marker, not *Mymk*. This choice was made because no commercially available antibody reliably detects *Mymk* protein. This clarification has been added to the Discussion.

To address the quantification comment, we attempted to use light-sheet microscopy on whole mount embryos stained with anti-MHC and anti-*Mymx*. Unfortunately, the resolution in the conditions tested was insufficient to distinguish individual nuclei within MHC+ cells. To resolve this problem, we quantified the immune-stained dissociated somite cultures from 4 different embryos. This allowed us to quantify the nuclei content of MHC+ cells. Our analysis revealed that approximately 90% of Mc2 cells are mono-nucleated, while only about 10% are bi-nucleated from cells isolated from E11.5 embryos. This quantification (shown below) has now been included in Figure S4.

In conclusion, the presence of a minority of bi-nucleated MHC+ cells lacking *Mymx* expression does not contradict our main conclusion. While the downregulation of *Mymx* post-fusion is possible, our data support that the downregulation largely occurs in mono-nucleated cells.

9. For Figure 3C, this heat map does not suggest a *Mymk*⁺/*Mymx*⁻ cluster. MC1 cluster is certainly enriched for *MYMX*, but it would be helpful to show the percentage of *Mymx* expression in the two clusters. Can the authors please provide violin plots for *mymx* and *myomaker* expression? The point sizes of the feature plots, seem to obscure positive *Mymx* cells in the MC2 cluster. Additionally, no relative expression color scaling for the feature plots, so are these all cells with at least one count of *Mymx* or *Mymk*? A violin plot will better highlight enrichment in both intensity of expression and frequency of expression of these two genes in these populations.

We agree with the reviewer that the heatmap alone does not fully capture the distribution and intensity of *Mymk* and *Mymx* expression across Mc1 and Mc2 clusters. To address the reviewer's comment, we have taken additional steps to clarify and strengthen our analysis. In term of transitional expression patterns, as noted by the reviewer, *Mymx* is enriched in the Mc1 cluster. As discussed above, we argue that the Mc1 and Mc2 represent transitional cell states during myogenic differentiation. Therefore, residual *Mymx* expression in some Mc2 cells is expected as part of the downregulation process. This interpretation is consistent with our immunofluorescence data, which show that *Mymx* protein is generally undetectable in MHC⁺ cell in embryo sections.

To directly address the reviewer's request, we now include violin plots for both *Mymx* and *Mymk* expression across clusters. As stated by the reviewer, these plots illustrate both the frequency and intensity of expression, highlighting the increased expression of *Mymx* in Mc1 and *Mymk* in both Mc1 and Mc2 cells. We have also updated the feature plots to include relative expression color scaling, which better reflects the expression gradients of *Mymk* and *Mymx*. These updated visualizations, shown below, have been added to Figure S4 and referenced in the main text to support our conclusion that Mc2 are largely *Mymx* negative while expressing *Mymk*. Note: Cluster 7 shares all its markers with Mc2 and lacks any distinct signature (as seen in the heatmap in Fig. 3C, where no unique marker cluster appears for it). This supports the interpretation that it is a technical artifact.

10. The authors assess *Mc1* and *Mc2* populations during secondary myogenesis, which they indicate takes place between E14.5 and E18.5. Are they defining secondary myogenesis strictly based on age of the embryo or the cell fusion events occurring in muscle where primary is fusion of myocytes to form a new myofiber and secondary is fusion of myocytes to myofibers?

We thank the reviewer for raising this point. In our study, we refer to secondary myogenesis in accordance with the widely accepted developmental timeline described in the literature, which places this phase between E14.5 and E18.5 in the mouse. We acknowledge that secondary myogenesis is also characterized by distinct cellular behaviors, notably the fusion of newly formed myocyte with pre-existing myofibers.

11. Line 318: I am worried about the interpretation that *Mc1* rapidly fuses with either an *Mc2* or a myofiber in proximity. This was state in relation to the *Mymx-Cre* data. For the *Mymx-Cre* allele, the idea that *Mc1* cells are not labelled at E11.5 due to a lag of *Cre* expression and recombination is possible. It's unclear why these data could be related to rapid fusion events. Aren't *Mc1* cells around during embryogenesis, so they don't rapidly fuse? Are the *Mc1* cells between E14.5 and E18.5 *Tom+*?

A similar comment was made by Reviewer 1. Please see our response to their Comment 3. We have conducted several new experiments to address this comment, and the new data is shown in Figure 5.

12. What are the data that a *Mc1* cell can't fuse with another *Mc1* cell? For the model in Figure 4G, I don't see any direct data that *Mc1* cells fuse to *Mc2* cells. I would strongly suggest the authors remove the model. If the down-regulation of *Mymx* is necessary for fusion as they predict why would this process be used at other times of fusion from E14-E18.

This is an excellent comment. We acknowledge that our data does not directly demonstrate that Mc1 are incapable of fusing with one another. This limitation is due to technical challenges in tracking Mc1-Mc1 fusion in vivo. In response to this valid concern, we have revised our working model to incorporate the possibility of both Mc1-Mc1 and Mc1-Mc2 fusion events. This update model reflects a more inclusive interpretation of the potential fusion dynamics during early embryogenesis. As a result, Figure 5G has been removed, and Figure 7 has been modified. In this model, we suggest that after the first myocyte-myocyte fusion step, bi-nucleated myofibers with downregulated *Mymx* are formed.

We also suggest that during the secondary myogenesis, Mc1 continue to fuse with the myofibers (Mymx-) to increase their size. At this point, the Mc2 did not disappear, but we suggest that they seeded the formation of myofibers during the primary myogenesis as explained in the discussion: “*This finding suggests the possibility that Mc1 terminally differentiate to form the Mc2 population, creating a building block that serves as the scaffold for subsequent fusion events with Mc1. In this model, Mc2 can be conceptualized as “pioneer myocytes”.*”

Thanks to this comment, we believe this updated model better accommodates the reviewer’s concerns and provides a more comprehensive framework for interpreting the observed cell states and fusion behaviors.

Second decision letter

MS ID#: dev.204771R1

MS TITLE: Differentially expressed fusogens specify myocyte states to drive myogenesis

AUTHORS: Jean-François Côté; Sarah Nahlé; Awais Javed; Loïck Joumier; Yacine Kherdjemil; Julie Sitolle; Konstantin Khetchoumian; Yash Parekh; Wojciech Krezel; Mohan Malleshaiah; Fabien Le Grand; Michel Cayouette

Dear Dr Côté,

I am happy to tell you that your manuscript has been accepted for publication in Development, pending our standard publication integrity checks.

Reviewer 1

The authors have been highly responsive to the critiques of both reviewers. I have no more concerns with the manuscript.

Reviewer 2

The authors have dealt with my concerns and I am supportive of publication.